# The risk of indoor sports and culture events for the transmission of COVID-19

Stefan Moritz [1,5✉], Cornelia Gottschick [2,5], Johannes Horn[2], Mario Popp[1], Susan Langer[2], Bianca Klee[2], Oliver Purschke[2], Michael Gekle[3], Angelika Ihling[1], Frank D. L. Zimmermann [4] & Rafael Mikolajczyk[2✉]

Nearly all mass gathering events worldwide were banned at the beginning of the COVID-19 pandemic, as they were suspected of presenting a considerable risk for the transmission of SARS-CoV-2. We investigated the risk of transmitting SARS-CoV-2 by droplets and aerosols during an experimental indoor mass gathering event under three different hygiene practices, and used the data in a simulation study to estimate the resulting burden of disease under conditions of controlled epidemics. Our results show that the mean number of measured direct contacts per visitor was nine persons and this can be reduced substantially by appropriate hygiene practices. A comparison of two versions of ventilation with different air exchange rates and different airflows found that the system which performed worst allowed a ten-fold increase in the number of individuals exposed to infectious aerosols. The overall burden of infections resulting from indoor mass gatherings depends largely on the quality of the ventilation system and the hygiene practices. Presuming an effective ventilation system, indoor mass gathering events with suitable hygiene practices have a very small, if any, effect on epidemic spread.

[1] Section of Clinical Infectious Diseases, University Hospital Halle (Saale), Halle, Germany. [2] Institute for Medical Epidemiology, Biometry and Informatics, PZG, Martin-Luther-University Halle-Wittenberg, Halle (Saale), Germany. [3] Julius Bernstein-Institute of Physiology, Faculty of Medicine, Martin Luther-University Halle-Wittenberg, Halle (Saale), Germany. [4] Zimmermann and Becker GmbH, Consulting Engineers, Dieselstr. 11, Flein, Germany. [5] These authors contributed equally: Stefan Moritz, Cornelia Gottschick. ✉email: Stefan.moritz@uk-halle.de; Rafael.mikolajczyk@uk-halle.de

In the course of the COVID-19 pandemic, banning mass gathering events (MGE) was one of the first countermeasures undertaken by the governments of most countries[1]. In Germany, early in March 2020, the government issued a general ban of MGE with more than 1000 people[2]. With a turnover of 129 billion Euro in 2019, the event sector is the sixth largest economic sector in Germany, and up to 1.5 million people depend on this industry[3]. Insolvencies in this field will not only have an economic impact, but may also result in the loss of creative skills, training infrastructure and a lack of upcoming young artists and athletes. The impact of this loss is not restricted to individuals, but affects an important dimension of society as a whole.

Severe acute respiratory syndrome coronavirus 2 (SARS-CoV-2), causing COVID-19, can be transmitted via droplets, aerosols or through contaminated surfaces[4–8]. While the debate on relevance of various transmission routes for the spread of COVID-19 is still ongoing[9–11], it is clear that physical proximity and hygiene determine transmission. Reported or measured personal contacts can be used to assess droplet based transmission. There are several additional factors of importance for studying aerosols. The type of activity and the resulting ex- and inhalation of emitters and recipients, as well as the airflow in the area around the recipients, must be taken into account[12–14].

To investigate the transmission risk of SARS-CoV-2 through droplets and aerosols during experimental indoor MGE, we conducted an experimental pop concert with three different hygiene practices, and measured the contacts of each spectator during the event using contact tracing devices (CTDs). We developed a computer model of the arena indoor space and simulated infectious aerosol distribution and the resulting exposure of healthy subjects. Finally, we combined information on contacts during the event and exposure to aerosols with an individual based model to estimate the excess burden of epidemic caused by indoor MGE. We incorporated various parameters, including the effects of different hygiene practices, wearing masks, event sizes, ventilation systems, and different baseline incidences in our model. We derived recommendations from this data regarding MGE during a pandemic.

## Results

### Contact measurements at the experimental mass gathering event.
We conducted an experimental pop concert on August 22$^{nd}$ 2020, with a total of 1212 individuals in the Leipzig Arena (Supplementary Table 1). All participants and involved staff demonstrated a negative test result for SARS-CoV-2, performed 48 h before the event. All people involved wore N95 masks during the event. Three different scenarios were investigated: 1) No restrictions (the pre-pandemic setting), 2) moderate restrictions (checkerboard pattern seating, twice as many entrances as in 1), 3) strong restrictions (pairwise seating with 1.5 m interspace to the next pair, four times as many entrances as in 1). Each scenario had the same schedule: first half, half time including simulated catering, second half, and exit. Contacts within a radius of 1.5 m were measured with a CTD.

There was a high overall number of contacts when all contacts over 10 s were counted. When considering only critical contacts with a duration of more than 15 min (based on the standard definition for contact tracing[15]) the number of contacts decreased below 10 (Fig. 1a, Supplementary Table 2). In each scenario a high number of contacts was observed during entry, half time and exit. Few of these contacts lasted more than 15 min during entry and half time (Fig. 1b). No contacts over 15 min were recorded in any of the scenarios in the exit phase. Few contacts were observed during the two halves, but nearly all lasted longer than 15 min.

The hygiene practices in Scenarios 2 and 3 resulted in a strong reduction in contacts of any duration. In Scenario 1, new contacts lasting longer than 5 min were created throughout the event, while in Scenarios 2 and 3 most contacts occurred during the entry phase, without further major increase (Fig. 1c). Overall, no effect of gender or age was observed regarding the number of contacts during the event (Supplementary Fig. 1).

### Simulation of aerosol exposure.
In addition to the number of contacts measured by CTD, the aerosol distribution in the respiratory air of all 4000 virtual participants was simulated using a computational fluid dynamics (CFD) model, considering two different ventilation versions (VV). Ventilation Version 1 (VV1) represented the current ventilation system in the arena. Here, the inlet air is blown in laterally on the east- and west side by jet nozzles (Supplementary Movie 1). Air supply was also issued under the seats of the grandstands through swirl diffusers, and below the mobile grandstands through ventilation grilles. The exhaust air was discharged in the corners of the arena by exhaust towers. Air exchange per hour (ACH) was 1.46 h$^{-1}$, with a make-up air of 50 m$^3$ h$^{-1}$-person. The make-up air is defined as the amount of air provided to a person in a room in one hour. To avoid large eddies (Supplementary Fig. 2a), which generate the intensified spread of aerosols at face level, jet nozzles and exhaust towers were turned off in Ventilation Version 2 (VV2) and the exhaust towers were replaced by exhaust pipes located under the roof, resulting in an ACH of 0.85 h$^{-1}$. This solution was chosen because its implementation would be cost-efficient, and in the hope that a displacement flow in the direction of the roof would be created by buoyancy induced flow. Unfortunately, the buoyancy induced flow was too weak due to the low occupancy and interfering air supply nearby the grandstands. Stationary eddies also emerged above the grandstands (Supplementary Fig. 2b). In the VV1 ventilation strategy, 24 infectious persons placed in the arena resulted in 85 individuals exposed to infectious aerosols, and the number for ventilation strategy VV2 was substantially higher (612 persons, Table 1). The overall mean exposure increased from 3.54 (VV1) to 25.50 exposed individuals (VV2), also resulting in a seven-fold increase in total. The mean exposure in the stalls was 6.75 (VV1) compared to 24.25 (VV2), resulting in an almost four-fold increase in exposed individuals. In the mobile grandstands, the mean exposure increased from 10.25 (VV1) to 59.75 (VV2), resulting in a six-fold increase. In the solid grandstands, the mean exposure changed from 4.25 (VV1) to 69.0 (VV2). Here, the increase was highest (16-fold), compared to the other areas (Table 1, Supplementary Fig. 3). Supplementary Fig. 4 shows the aerosol concentration displayed as isosurfaces around the infectious individuals. The isosurfaces show the same transmission mechanism for both VV1 and VV2, irrespective of the position: direct aerosol flow from the mouth of the transmitting individual to the mouth of the recipient. Differences in the number of infected individuals between the two ventilation variants can be explained by the lower air exchange per hour (ACH) of 0.85 h$^{-1}$, as well as less air movement (and therefore slower mixing of the air) in VV2.

The estimated mean number of exposed people per one infectious person was 3.5 (±2.9 standard deviation (SD)) in VV1, and 25.5 (±27.8 SD) in VV2 for Scenario 1, with a maximum of 10 and 108 exposed persons respectively (Supplementary Fig. 5). Hygiene practices reduced the mean number of exposed visitors in both VV1 and VV2, to 1.9 (±1.5 SD) and 11.8 (±13.5 SD) for Scenario 2 and to 0.7 (±1.0 SD) and 5.3 (±6.4 SD) for Scenario 3.

### Effect of mass gathering events on SARS-CoV-2 positive cases in the population.
In order to assess transmissions in indoor

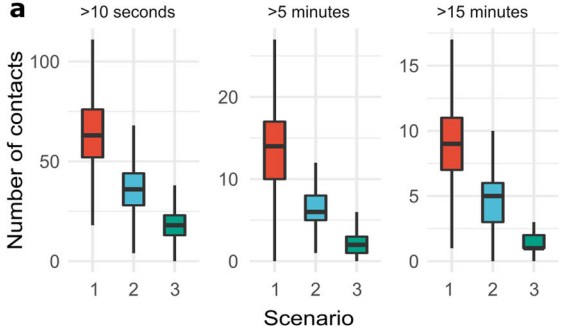

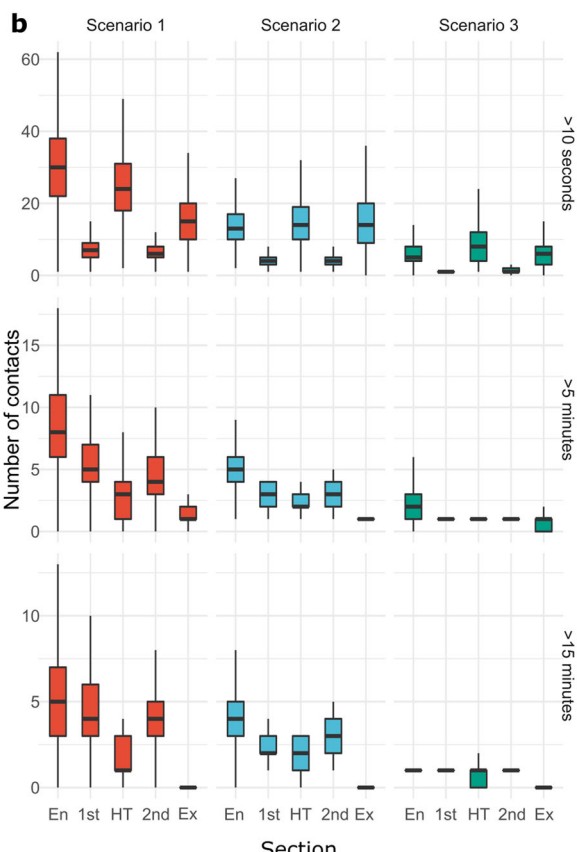

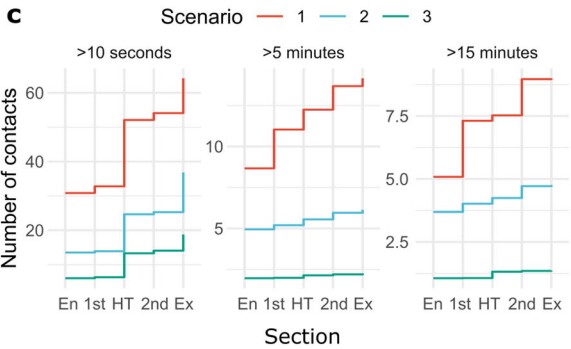

**Fig. 1 Number of contacts in Scenario 1 (red), 2 (blue), and 3 (green). a** Number of contacts by duration (>10 s, >5 min, >15 min) over all settings, **b** for the different sections: Entry (En), 1st half (1st), half time (HT), 2nd half (2nd), exit (Ex) and **c** cumulatively throughout the different settings. The center line represents the median, the box limits the upper and lower quartiles and whiskers extend from the hinge to the smallest/largest value no further than 1.5 * IQR from the hinge. $n_{Scenario\ 1} = 1192$ participants, $n_{Scenario\ 2} = 1158$ participants, $n_{Scenario\ 3} = 1054$ participants. See Supplementary Table 3 for components of the hygiene practices in the three different scenarios.

consequence incidence is independent of the local transmission dynamics represented by the reproduction number. In line with findings from serological studies, we assumed seroprevalence to be still negligible, making most people susceptible to infection[16–20].

For certain predefined incidences (based on diagnosed cases), we simulated the number of infectious persons attending the MGE. Per definition, these persons were not diagnosed yet, and displayed no symptoms at the time of the MGE—otherwise they would not attend the event. Similarly, quarantined persons are excluded, and (asymptomatic) cases identified by contact tracing could not attend the event. Conversely, undetected asymptomatic, pre-symptomatic and some fraction of mildly symptomatic persons could attend the event. Some of these persons would remain undiagnosed and contribute to the dark figure of cases. The testing strategy and control measures resulted in a controlled epidemic, in which about 50% of the infected were detected during the course of infection. The number of infectious MGE participants was obtained from the agent-based model accounting for these aspects.

Depending on the incidence (i.e. 10, 50, or 100 positive tested cases per 100,000 population per week), on average 7.8, 37.8, or 75 infectious persons might attend any event, assuming the total number of persons taking part in MGE is 200,000 per month (corresponding to the number of participants in MGE in Leipzig in pre-pandemic times, Supplementary Fig. 6). We assumed a transmission probability of about 7% per contact of 15+ min duration based on German POLYMOD data[21], and a reproduction number of around 1 without any contact restrictions. The resulting additional average numbers of persons who would become infected and would be detected (excess cases) ranges from 5.1 under the strictest hygiene practice and best ventilation (Scenario 3, VV1) to 22.0 with no hygiene practice and non-optimal ventilation (Scenario 1, VV2) in the low incidence scenario (10 per 100,000 per week) and with spectators wearing masks (Table 2). An increased incidence of 100/100,000/week results in 11.7 and 196.8 persons likely to acquire an infection during an MGE for the same conditions. Not wearing masks further increases these numbers (Supplementary Data 1). One hospitalization and possible death due to MGE could occur at an incidence of 50/100,000/week, assuming bad ventilation (Supplementary Data 1). At an incidence of 100/100,000/week, hospitalized cases and deaths could occur in both VVs. The number of cases are respectively lower for MGE with an overall size of 100,000 participants per month, and higher for MGE without mask wearing (Supplementary Data 1). For MGE with 200,000 participants, there is a 23.6%, 10.8%, and 4.5% increase of positive cases attributed to MGE for Scenarios 1, 2, and 3 without the use of masks in VV2, which decreases to 4.8%, 2.8%, and 1.2% with masks and better ventilation (VV1) in the high incidence setting (100 per 100,000 per week, Table 3). These numbers can again be reduced, assuming smaller event sizes (Supplementary Data 2). This highlights the importance of adequate ventilation

MGEs, we developed a dedicated individual-based model. We investigated the effects of MGE in epidemics controlled through the overall reduction of contacts within society and contact tracing of SARS-CoV-2 positive individuals, mimicking a situation in Germany during summer 2020 with a reproduction number ~1. We allowed the introduction of new cases (i.e. by persons visiting from regions with a higher incidence), and in

**Table 1 Differences in overall and mean exposure to infectious aerosols in ventilation version 1 (VV1) and ventilation version 2 (VV2) for Sscenario 1.**

|  | Infectious | VV1 | | VV2 | |
| --- | --- | --- | --- | --- | --- |
|  |  | Exposed | Mean [Exposed] | Exposed | Mean [Exposed] |
| Stalls |  |  |  |  |  |
| P1 | 2 | 8 | 6.75 | 24 | 24.25 |
| P2 | 2 | 13 |  | 33 |  |
| P7 | 2 | 2 |  | 13 |  |
| P8 | 2 | 4 |  | 27 |  |
| Mobile Grandstands |  |  |  |  |  |
| WM2 | 3 | 15 | 10.25 | 89 | 59.75 |
| WM5 | 3 | 9 |  | 40 |  |
| OM2 | 3 | 8 |  | 50 |  |
| OM5 | 3 | 9 |  | 60 |  |
| Solid Grandstands |  |  |  |  |  |
| WF2 | 1 | 5 | 4.25 | 55 | 69.00 |
| WF5 | 1 | 9 |  | 45 |  |
| OF2 | 1 | 2 |  | 68 |  |
| OF5 | 1 | 1 |  | 108 |  |
| Sum | 24 | 85 |  | 612 |  |
| Mean Exposed Total |  |  | 3.54 |  | 25.50 |

P1, P2, P7, P8—Locations of infectious individuals in the stalls; WM2, WM5, OM2, OM5—Location of infectious individuals in the mobile grandstands; WF2, WF5, OF2, OF5—Location of infectious individuals in the solid grandstands. Infectious: number of infectious individuals, exposed: number of exposed individuals. WF: upper west grandstand; WM: lower west grandstand; OF: upper east grandstand; OM: lower east grandstand; P: floor. See Supplementary Fig. 8 for a map of the arena.

**Table 2 Simulated mean excess numbers caused by mass gathering events (MGE).**

| IN | S | PC | AC | LC | SC | H | Q |
| --- | --- | --- | --- | --- | --- | --- | --- |
| Ventilation Version 1 |  |  |  |  |  |  |  |
| 10 | 1 | 11.5 [−110.4; 129.2] | 1.9 [−20.0; 26.0] | 7.8 [−82.0; 94.2] | 1.7 [−19.0; 21.0] | 0.3 [−6.0; 6.0] | 19.5 [−222.2; 279.2] |
|  | 2 | 8.5 [−100.4; 120.2] | 1.3 [−20.0; 24.0] | 6.0 [−78.2; 91.2] | 1.2 [−18.0; 21.0] | 0.2 [−6.0; 6.0] | 14.4 [−214.4; 258.2] |
|  | 3 | 5.1 [−107.2; 115.0] | 1.1 [−20.2; 26.0] | 3.4 [−81.2; 86.2] | 0.6 [−19.0; 21.0] | 0.0 [−6.0; 6.0] | 8.7 [−233.0; 252.4] |
| 50 | 1 | 33.8 [−203.2; 277.2] | 6.9 [−43.2; 56.2] | 20.8 [−151.6; 197.0] | 6.1 [−33.0; 47.0] | 1.0 [−12.2; 14.0] | 50.4 [−440.2; 600.0] |
|  | 2 | 21.8 [−221.2; 272.4] | 4.7 [−44.0; 57.0] | 13.6 [−163.0; 189.0] | 3.6 [−38.2; 47.0] | 0.8 [−12.0; 14.0] | 33.8 [−456.0; 543.8] |
|  | 3 | 1.9 [−251.4; 253.4] | 1.0 [−48.2; 50.0] | 0.3 [−185.2; 181.6] | 0.6 [−47.2; 43.0] | −0.1 [−13.0; 13.0] | 2.1 [−531.2; 533.4] |
| 100 | 1 | 71.4 [−258.6; 422.0] | 11.3 [−57.0; 81.2] | 46.9 [−194.4; 306.0] | 13.1 [−45.2; 71.0] | 2.5 [−16.0; 20.0] | 124.3 [−632.6; 837.2] |
|  | 2 | 37.3 [−301.6; 409.0] | 5.9 [−60.2; 70.2] | 24.2 [−230.4; 287.4] | 7.2 [−53.0; 65.2] | 1.2 [−17.0; 20.0] | 50.6 [−718.0; 777.4] |
|  | 3 | 11.7 [−330.2; 353.6] | 1.3 [−68.0; 72.4] | 8.9 [−247.0; 256.6] | 1.5 [−57.2; 55.2] | 0.4 [−17.0; 18.0] | 28.5 [−730.0; 773.0] |
| Ventilation Version 2 |  |  |  |  |  |  |  |
| 10 | 1 | 22.0 [−94.4; 147.2] | 3.8 [−21.0; 27.0] | 14.0 [−73.2; 105.0] | 4.2 [−16.0; 25.0] | 0.8 [−5.0; 7.0] | 36.9 [−220.2; 300.4] |
|  | 2 | 9.1 [−106.2; 126.6] | 1.4 [−22.0; 24.0] | 5.7 [−78.2; 90.2] | 1.9 [−18.0; 21.2] | 0.4 [−5.0; 7.0] | 12.7 [−227.8; 260.4] |
|  | 3 | 4.4 [−104.2; 114.2] | 0.7 [−22.2; 22.2] | 2.8 [−77.2; 85.0] | 0.8 [−18.0; 19.0] | 0.2 [−5.0; 6.0] | 7.8 [−238.0; 245.2] |
| 50 | 1 | 99.4 [−152.6; 348.8] | 16.8 [−35.2; 71.0] | 63.7 [−112.2; 243.0] | 18.8 [−24.0; 66.0] | 3.5 [−10.0; 17.0] | 160.4 [−352.6; 694.4] |
|  | 2 | 47.6 [−189.0; 295.0] | 7.8 [−43.0; 58.2] | 30.9 [−140.2; 211.2] | 8.8 [−32.2; 53.0] | 1.7 [−11.0; 14.0] | 82.2 [−416.0; 605.2] |
|  | 3 | 10.3 [−223.4; 248.4] | 2.0 [−43.0; 52.0] | 6.1 [−160.0; 185.2] | 2.3 [−37.4; 42.0] | 0.7 [−12.0; 13.0] | 22.0 [−513.0; 523.2] |
| 100 | 1 | 196.8 [−170.6; 558.8] | 34.1 [−37.2; 108.2] | 124.0 [−146.2; 392.2] | 38.6 [−22.0; 102.0] | 7.6 [−11.0; 27.0] | 329.5 [−423.2; 1084.0] |
|  | 2 | 96.9 [−233.2; 425.4] | 16.4 [−51.2; 83.0] | 62.2 [−181.2; 299.0] | 18.3 [−38.2; 76.2] | 3.3 [−14.2; 22.0] | 170.7 [−500.2; 865.8] |
|  | 3 | 30.5 [−294.4; 343.2] | 5.1 [−59.0; 73.0] | 19.0 [−224.0; 251.6] | 6.4 [−49.2; 62.0] | 1.2 [−16.0; 19.0] | 58.8 [−658.6; 752.8] |

Mean excess numbers of overall positive cases (PC), asymptomatic cases (AC), light symptomatic cases (LC), severe symptomatic cases (SC), hospitalized cases (H), and quarantined individuals (Q) for each ventilation version (VV) by background incidence (IN) per 100,000 per week and scenario (S), with MGE attended by 200,000 individuals per 30 days. Mask wearing is assumed. Empirical 95% confidence intervals calculated from all runs are shown in brackets.

for the reduction of transmission risk and demonstrates that mask wearing and event size adjustments are possible measures for risk reduction.

**Acceptance of hygiene practices**. Acceptance and compliance are key factors in achieving the full potential of hygiene practices regarding contact and transmission reduction. A total of 960 study participants completed a questionnaire provided three weeks after the experiment (79%). Of those, 88% could imagine attending an event or concert under the conditions of Scenario 2, and 82% under the conditions of Scenario 3. The majority of

respondents (89%) felt the wearing of a N95-mask was unproblematic, or a little restrictive, but that they could get used to it quickly (Supplementary Fig. 7). If it were necessary to wear normal mouth-nose protection or a N95 mask for a concert, 90% and 78% of the participants, respectively, would do so.

## Discussion
Even without precautions, not every attendant has contact with all others during an MGE. In scenarios with physical distancing, the resulting contact numbers are low and the effective risk depends primarily on the adequacy of the ventilation. Under

**Table 3 Increase in SARS-CoV-2 positive cases in percentage that acquired the infection during the mass gathering events (MGE).**

| | IN | S | Increase of SARS-CoV-2 positive cases [%] | |
|---|---|---|---|---|
| | | | No masks | Masks |
| Ventilation Version 1 | 10 | 1 | 13.3 [-43.7; 112.8] | 13.3 [-45.4; 115.3] |
| | | 2 | 11.3 [-46.8; 120.8] | 11.6 [-45.3; 109.2] |
| | | 3 | 7.7 [-45.9; 97.5] | 9.2 [-46.9; 96.8] |
| | 50 | 1 | 9.2 [-19.7; 39.9] | 5.0 [-20.5; 35.8] |
| | | 2 | 5.1 [-21.0; 35.6] | 3.7 [-23.2; 38.0] |
| | | 3 | 2.6 [-22.9; 31.9] | 1.4 [-26.7; 34.4] |
| | 100 | 1 | 9.1 [-11.1; 30.6] | 4.8 [-14.2; 27.4] |
| | | 2 | 4.8 [-14.4; 28.6] | 2.8 [-16.7; 26.7] |
| | | 3 | 2.3 [-17.3; 25.0] | 1.2 [-17.6; 22.5] |
| Ventilation Version 2 | 10 | 1 | 29.2 [-40.3; 136.8] | 18.7 [-43.5; 114.2] |
| | | 2 | 15.6 [-44.2; 113.0] | 11.0 [-47.2; 97.6] |
| | | 3 | 11.2 [-47.0; 104.9] | 8.3 [-47.3; 93.9] |
| | 50 | 1 | 24.6 [-8.1; 64.7] | 12.6 [-15.4; 45.0] |
| | | 2 | 11.7 [-16.1; 45.5] | 6.5 [-20.5; 39.7] |
| | | 3 | 5.3 [-21.8; 36.5] | 2.2 [-23.0; 33.1] |
| | 100 | 1 | 23.6 [-0.2; 49.9] | 12.2 [-10.1; 36.3] |
| | | 2 | 10.8 [-9.7; 35.2] | 6.2 [-12.3; 27.8] |
| | | 3 | 4.5 [-14.5; 25.5] | 2.3 [-16.2; 24.2] |

At an incidence of 10 per 100,000 people, random effects have a strong impact on the number of additional cases due to MGE, such that the variation supersedes the impact of masks. The mean across all 1000 runs is therefore affected by stochastic fluctuation.
Numbers of positive cases are shown for MGE with 200,000 participants per month, and by scenario 1-3 compared to no MGE, including mask wearing vs. no masks, ventilation version 1 and 2, and different incidences per 100,000 per week (IN). The increase in positive cases was calculated by dividing the number of cases with MGE minus the number of cases without MGE divided by the number of cases without MGE, thus negative numbers indicate that a simulation run without MGE had higher numbers compared to the parallel run with MGE. Empirical 95% confidence intervals calculated from all runs are shown in brackets.

hygiene protocols and with good ventilation, even a substantial number of indoor MGE would thus have minimal effects on the overall number of infections in the population. However, poor ventilation systems can lead to a considerably higher rate of aerosol exposure, and can thereby result in a high number of infections. MGE only contribute to a small proportion of all individual contacts within a population.

In our simulations the difference with and without events was close to zero on average, but in some cases, the numbers of new infections could be substantial. In an unfavorable case, this may result in the impression that many infections were caused by events. Apart from these single unfortunate MGE, events without any precautions can make a substantial contribution to the epidemic spread. Under precautionary measures, MGE will contribute only a small fraction of new cases to the overall epidemic, even at R values above 1. Some contacts might also not be truly additional contacts from MGE, as people attending the event may have been previous contacts, meaning the overall effect of MGE on transmission is further diminished.

While poor ventilation can substantially increase the number of transmissions, we expect that using masks, and particularly N95 masks, would reduce the risk. Masks were not used in the super-spreading events described in the literature[22]. The effects of masks on the reduction of transmission are generally accepted[23,24]. We compared MGE with and without mask wearing, and, since the effects are proportional within an MGE, the effect of wearing masks by any percentage of the participants can be directly estimated. Nevertheless, aerosols are of special concern in indoor settings, during the periods when visitors stay

in their seats, and exposure time can accumulate. Wearing masks should therefore be mandatory, especially while sitting, to maximize its protective effect. The entry, half time and exit phases are important with respect to contacts, but particularly for uncritical short time contacts.

Hygiene practices must thus address organizational aspects to ensure low contact times in all periods. Testing before an event is likely to give additional security, but it is very time and resource consuming. In addition, testing thousands of people within a few hours would be a huge organizational challenge. Our results apply to MGE with seating orders and a high compliance with the implemented hygiene practice. Hygiene stewards rarely had to intervene in our experiment. This might be a consequence of the highly disciplined participants in our study, but also indicates that knowledge about being tested negatively did not lead to the breach of specific distancing rules in the various phases of the experiment. Enforcing a hygiene practice in routine practice is crucial for risk reduction, however, and can be supported by hygiene stewards.

Large scale events (e.g. soccer games) and standing concerts (e.g. rock concerts) might be different to the MGE we simulated with respect to the number of contacts and the probability of transmission[23]. Primarily, larger crowds cause people to stand closer together during entry due to space restrictions, causing additional contacts on the way to the event. Secondly, especially in unseated concerts, visitors are in a close proximity to each other, and do not stay in fixed positions, so the number of contacts can increase over time.

There are several limitations to our study. First, we did not reach our intended goal of 4000 participants. Although we implemented space restrictions, the density of contacts may still have been reduced. Second, we made simplifications regarding aerosol exposure, as crucial aspects such as the minimal infectious dose or the viral load of aerosols remain unknown. The ASHRAE Standard 62.1-2013 recommends a minimum ventilation rate of $3.8\,l\,s^{-1}$ per person in the spectator area. Due to an air supply of $198\,000\,m^3\,h^{-1}$ the Leipzig arena supplies $6.7\,l\,s^{-1}$ per person (max. 8200 seating persons) or $4.5\,l\,s^{-1}$ (max. 12,300 standing persons), and exceeds the ASHRAE recommendation by approximately 1.75 times or 1.2 times, respectively. Since ventilation is crucial for the risk associated with MGE, it is important that further studies focus on this aspect. Third, we did not analyse other opportunities for contacts which could be linked to the MGE. For example, additional contacts could take place during transit, or if participants of the MGE go to bars or similar venues after the concert. We assumed that all the other settings would have their own hygiene practices, for example, not allowing overcrowding. In a practical sense, this can be difficult for large events, but with additional efforts the excess risk can be minimized. While we used a detailed model to simulate SARS-CoV-2 transmission in society, additional structures in the population can affect the results. For example, if the same group participated in all events and transmitted the infections acquired in one event to another, it would result in a higher impact of MGE. Fourth, lack of adherence to hygiene practices is a potential danger, but reinforcing the hygiene practice should be a requirement for MGE. Lastly, our model assumes no limit for contact tracing capacities. If health authorities were overloaded and contact tracing compromised, the effect of MGE would be higher than assumed in our model. During the early pandemic in Germany, an incidence of 50 per 100,000 per week was considered the threshold below which contact tracing capacities were sufficient. Contact tracing capacities were later expanded, but still it is possible that the capacities would be insufficient for higher incidences.

In conclusion, we found that the participants of a seated concert in a well-ventilated arena have a high number of short

contacts and a low number of long lasting contacts. A moderately restrictive hygiene practice (i.e. Scenario 2) provided a substantial reduction in infection risk. Wearing masks during the concert was highly accepted by most participants and can provide further risk reduction. When hygiene practices are applied and the conditions of good ventilation are met, MGE appear to contribute little to the epidemic spread of COVID-19. A lack of hygiene practices and/or inadequate ventilation can substantially increase the number of subjects at risk.

## Methods

**General study design**. The "Risk Prediction of Indoor Sports and Culture Events for the Transmission of Covid-19" (RESTART-19) study was initiated in order to provide data on contacts and aerosol exposure at indoor mass gathering events (MGE). The study comprises three parts:

1. Experiment: In order to determine the number of contacts during a MGE, we conducted a pop concert under experimental conditions and provided all participants with a contact tracing device (CTD). The concert was performed in three scenarios with different hygiene practices.
2. Aerosol Distribution: To assess the aerosol exposure, the aerosol movement, the indoor aerosol concentrations and the concentrations in the breathing air were calculated using computational fluid dynamics (CFD).
3. Epidemiological Simulation: We integrated the results of contact tracing and aerosol distribution in an individual-based model, and simulated the effects on the subsequent burden of infections.

The study protocol was submitted to the German clinical trial register (DRKS 00022790). (www.drks.de)

**Experimental concert simulation and contact measurement**. The event took place on August 22nd, 2020 in an indoor arena (Quarterback Immobilien Arena, QIA) in the city of Leipzig (Germany).

*Recruitment procedure and participants*

Individuals aged between 18 and 50 years were invited through an extensive media campaign to register voluntarily and free of charge via the study webpage (www.restart19.de), where comprehensive information about the event, its objectives, and risks was provided. All participants gave their informed consent. Participants did not receive any kind of allowance, but food and drinks were provided for free throughout the experiment. A priori exclusion criteria were self-reported obesity (Body-Mass-Index >30), chronic diseases, cardiovascular diseases, cancer, immune suppression, the intake of immunosuppressants, pregnancy, or conditions affecting lungs, liver, or kidneys. We planned to include 4000 participants, corresponding to half of the capacity of the arena, and reflecting the mean event size for sports and culture events in 2019 at this location (4200 participants).

From July 17th to August 21st, 2020 a total of 2825 participants registered for the study. 601 participants actively withdrew their consent, and 212 participants did not confirm their registration. 2023 participants thus received the SARS-CoV-2 screening test set (see hygiene practice below), of which 1407 samples were returned on time to the laboratory for analysis. Information on test results was reported back to the participants. Only those with a negative test result were asked to participate. One participant tested positive and was therefore excluded. In total, 1212 persons took part in the experiment. Because fewer individuals participated in the event than initially planned, we prepared the arena to create a setting of realistic density: we closed seating ranks, catering stalls, bathrooms, and entries as required by each hygiene practice. In Scenario 1, 1192 participants, in Scenario 2, 1158 participants and in Scenario 3, 1054 participants were present. These numbers are smaller than the total number of participants ($n = 1212$), because not all participants were present during all scenarios.

**The event**. On the study day, all participants arrived between 8:00 and 10:00 a.m. for check-in. During check-in, participants were registered, identities confirmed and N95 masks, hand sanitizers, and contact tracing devices handed out to each person. Three tickets for three different scenarios were issued per person, containing information on timing, entrances, and seating for each scenario.

We simulated three different scenarios in order to analyse the impact of different hygiene measures on the transmission of SARS-CoV-2. Each scenario followed the same schedule: entry (60 min), 1st half (20 min), half time (20 min), 2nd half (20 min), exit (15 min). During the halves, the German singer/songwriter Tim Bendzko performed a live pop concert. Scenarios differed with respect to hygiene measures such as number of entrances/exits, distance between seats, and restricted mixing of participants by dividing the arena into quadrants. Scenario 1 was designed to reflect a pre-pandemic state, where participants entered and exited the arena through two main entrances without any restrictions, and were seated without free seats in-between. Scenario 2 applied moderate hygiene measures: the arena was divided into four quadrants. Participants entered and exited the arena through the entrance/exit of the quadrant as indicated on their ticket (four

entrances/exits) and were not allowed to change quadrants. A seating arrangement was implemented, where every second seat was occupied and the rows were shifted (checkerboard pattern). Scenario 3 reflected a stronger contact reduction, with pairwise seating of participants and the implementation of a minimum distance of 1.5 m between the occupied pairs of seats. The number of entrances/exits was also increased to eight. The different scenarios are summarized in Supplementary Table 3.

**The setting**. The Quarterback Immobilien Arena is an event location in the city of Leipzig and is one of the 10 most frequented live entertainment venues in Germany (https://www.stadionwelt.de/plus/arena-ranking-besucher). The type and layout of the arena (multipurpose hall) is common in the industry, and further examples can be found in Stuttgart (Porsche Arena), Berlin (Max Schmeling Arena) and Nuremberg (Arena Nuremberg Insurance). The arena has a seating capacity of up to 8228 people. Supplementary Fig. 8 shows an overall plan of the location. Visitors usually (i.e. before the pandemic) enter the hall via two main entrances (west and east side) opening into the foyer at the south end of the arena. From the foyer, they enter two long tunnels running parallel to the interior on each side of the hall. Visitors reach the event room via corridors branching off the tunnels. The arena also has four emergency exits on each long side of the building, which were used in Scenarios 2 and 3 to enter and exit the arena.

The total room volume of the arena is 135,000 $m^3$. The ventilation system has a total capacity of 198,000 $m^3 h^{-1}$ and uses 100% fresh air. The outlets under the grandstand have a capacity of 114 000 $m^3 h^{-1}$. On the long side of the grandstand, there are jet nozzles above the heads of the spectators, which blow air downstream to the inner space (84.000 $m^3 h^{-1}$).

**Hygiene practice**. The Saxonian Ministry of Social Affairs and Cohesion (Sächsisches Staatsministerium für Soziales und Gesellschaftlichen Zusammenhalt) and the health authorities of the city of Leipzig approved the hygiene practice.

SARS-CoV-2 testing: One week before the event, all participants and staff members received a PCR test set for SARS-CoV-2 including a swab and a tube containing stabilizing solution. The set included detailed instructions on self-sampling and on returning the test set. Participants were requested to take a throat swab within the 48 h before the event. The test sets could be returned to five different locations in Leipzig or Halle (Saale), or sent via mail. All samples were analyzed by the Institute of Virology of the University Hospital in Leipzig. Test results were imported to the data bank the night before the event and the participants received notification via e-mail. Participants with positive or missing test results were informed by phone, and were not allowed to enter the arena. The test was free of charge.

Exclusion criteria: Onsite exclusion criteria were no valid registration, no ID, a positive or missing SARS-CoV-2 test, temperature above 37.5 °C, self-reported symptoms of COVID-19 within the past 48 h, contact with a COVID-19 patient or a stay in a risk area (according to the Robert Koch Institute of August, 22th 2020) within the last 14 days.

Personal protective equipment: During study-check-in, each participant received a N95 mask, a hand sanitizer bottle containing 85.5% Ethanol V/V, as well as an ultra-wide band contact tracing device (CTD). The N95 mask had to be worn from entering to leaving the arena, as well as outside the arena in queues at the entrance and exits.

Catering: A catering service was only provided outside the arena, where participants were allowed to remove their masks if a distance of 1.5 m could be maintained. The catering was free of charge in order to reduce the waiting time for participants and to avoid participants leaving the area. Water bottles were given to the participants on request inside the arena, where they were allowed to drink while maintaining an appropriate distance from other people. Indoor catering service was simulated during the half times of the different scenarios, so that people received vouchers to use outside.

Distance and hygiene stewards: Except for the first scenario, all participants were asked to keep a distance of 1.5 m. To ensure that all participants followed the hygiene practice, 40 hygiene stewards were present inside the arena. Participants repeatedly not adhering to the hygiene practice advice would have been asked to leave the arena (but this was not necessary).

Briefing of participants and staff: All participants received comprehensive information regarding the hygiene practice upon registration. On the day of the event, the participants received an information sheet with the hygiene rules and instructions on proper use of N95 masks and hand sanitizers by the check-in staff. Participants also received verbal instructions at the beginning of the event. Staff received detailed training on hygiene practices.

Contact tracing: Participants provided full contact details during the registration process. Participants agreed that their contact tracing devices (CTD) could be used to identify those at risk. In case of a SARS-CoV-2 infection after the event, affected participants would have been contacted. We are not aware of any persons who were infected during the event. All personal information was deleted six weeks after the event. The resulting data was anonymized.

Corona Warn App: Using the Corona-Warn-App of the German federal government was recommended, but not required for participation.

**Measurement of contacts in physical proximity.** All participants received a personalized contact tracing device (CTD) and were instructed to wear it around their neck during the event. ICDWpro quad 164643 (In-Circuit, Dresden, Germany) tags were used to measure the distance between two close participants and the contact duration time at this distance. These tags combine Bluetooth low energy and ultra-wide band radio technology, reaching an accuracy of ± 20 cm. The CTD could either send or receive signals at any given time point so that there was an exchange of signals between the CTDs of all participants. Distances were not measured constantly, but at specified intervals (around every three seconds). The tags coordinate their broadcast time to minimize interference (only distant tags broadcast at the same time), and therefore, very short encounters can be missed (<3 s), but longer encounters are recorded. The firmware and logging were customized according to the following protocol: a time stamp was recorded for the beginning of a contact if one of the following combinations of distance and time were met: <50 cm for at least 3 s, <100 cm for at least 6 s or <150 cm for at least 10 s. When leaving these thresholds for more than 2 s a time stamp was logged. When the contact was broken, a new contact could be recorded after a 10 s reset time.

The signals were often interrupted due to the high number of tracers sending data simultaneously, a low broadcast intensity and small changes in the distance movements of participants. A CTD could also only receive or send signals, but not do both simultaneously. For a pair of sensors, proximity was thus recorded partly on the one and partly on the other. First, contacts from all devices were combined. Second, gaps were filled in between the first and last contact within a phase of the scenario (for example during half time). Given this specification of the sensors, we were only able to use the information on the largest distance, i.e. 1.5 m (likely to correspond to a physical distance of 1.3 m when the sensors faced each other, and less when signals were partly obscured by body parts). For a more realistic assessment of contacts, we scaled up the halves to 45 min each. In this way, contacts with persons moving in and out of the radius of 1.5 during the sitting period could also accumulate and cross the threshold of 15 min (Supplementary Table 4). We studied the total number of contacts lasting >10 sec (we included contacts of >3 s for the distance of 50 cm and >6 for the distance of 1 m in this category) and 5 and 15 min. A critical contact was defined as lasting longer than 15 min within a distance of 1.5 m, in line with contact definition by the Robert Koch Institute. The 15 min could accumulate through the full event for the total number of contacts. Contact analysis was performed using R (version 4.0.2).

**Acceptability questionnaire.** All participants of the experiment were asked via email to complete an online survey three weeks after the event. The questionnaire contained 10 questions regarding perceptions and opinions of the feasibility of such an event. The focus of interest was on wearing masks and personal risk perception in the different scenarios.

**Aerosol distribution**
*Computational fluid dynamics (CFD).* Aerosol distribution within the arena was simulated using a computational fluid dynamics model. All CFD simulations were conducted with the commercial CFD software PHOENICS (Version 2020, CHAM, London, United Kingdom). For aerosol distribution the PHOENICS add-on FLAIR (drift-flux modelling), and for particle tracking the add-on GENTRA were used (both Version 2020, CHAM, London, United Kingdom). The PHOENICS program was developed by Professor Brian Spalding, and has been successfully used for more than 30 years by Zimmermann and Becker GmbH, Consulting Engineers (Germany) for the validation of technical planning, in a wide range of technical applications, and is validated for new applications. The PHOENICS/ FLAIR/ GENTRA model has been used in numerous other studies focusing on the CFD simulation of aerosols[25–28]. The Quarterback Immobilien Arena was exactly transferred into a 3D model (1:1), including all built-on components, the complete ventilation system, grandstands and seats. Virtual spectators were seated within the arena to simulate the aerosol emission and exposure. The position of the infectious persons was determined according to a pre-calculation for the analysis of the room air flow (Supplementary Fig. 2a). Further pre-calculations were carried out in a simplified model for the analysis of aerosol movement. These calculations showed that the trajectory was directly dependent on room air flow from 0.05 m s⁻¹ air velocity for aerosols with d ≤ 10 μm as well as for CO₂. Since the flow pattern is almost identical from the first row of stalls to the last, infectious persons were placed in the front and in the back of the stalls in such a way that the different room air flows in the stalls and on the stands were recorded. Due to the size of the model and the required accuracy of the grid, very long computation times were expected. We decided to model Scenario 2 and to interpolate the other scenarios from this data. We installed 4000 virtual spectators within the model with a seating arrangement following a checkerboard pattern (i.e. every second chair remains free). Twenty-four infectious persons were seated in 12 (out of 32) blocks. The detailed distribution of the infectious persons can be seen in Supplementary Fig. 8. The breathing air of all virtual spectators consists of an ideal gas and 20 litres CO₂ per hour. Infectious spectators also emit aerosols of different sizes (0.5, 5, and 10 μm) into the room while breathing. Emission rate and particle size were adjusted to that of singing people[29,30]. The assumed physical density of respiratory air was 1.3 g/ml[31]. We used a slightly increased value of 12 l min⁻¹ per person for the breathing volume and the corresponding proportion of CO₂, as we expected an increase during singing, shouting, or cheering. The detailed parameters and equations used for the model are summarized in Supplementary Tables 5 and 6. Each dummy was equipped with a virtual mouth for respiration to quantify the virus exposure of the spectators. Aerosol exposure was measured within the model directly at the mouth opening. To evaluate the dynamics and flow of the aerosol distribution within the arena, their aerosol movement was calculated via the particle tracking software GENTRA by the drift-flux model in FLAIR, and the results were presented through visualization by GENTRA and cumulated numbers. The quantitative results of the dispersion of the exhaled aerosols of the infectious participants were transferred to spreadsheets according to the seating arrangement in the arena. The number of affected persons and their level of exposure were obtained from these spreadsheets. We calculated an average value of persons exposed, in addition to those who would be captured by contact measurement with CTD due to critical contacts (Supplementary Table 4). In order to visualize the aerosol distribution, values below the critical threshold of $1.75 \times 10^{-3}\ \mu g\ s^{-1}$ aerosol exposure (Supplementary Fig. 5) were colored in green, and all values above were red. The brightness of the red color corresponds to the relative size of the mass flow.

*Variants of ventilation.* We ran the ventilation model in two different variants representing different ventilation systems. In the first ventilation variant, we modeled the actual ventilation system of the arena. Here, air reaches the arena via the outlets under the lateral grandstands and the jet nozzles, as described above. Two towers are installed in each corner of the hall for air suction. The air supply was 198,000 m³ h⁻¹, corresponding to an air exchange rate of 1.46 air changes per hour (ACH). In a second variant, we tried to modify the ventilation system in the arena to enable displacement flow by virtually installing two long exhaust pipes under the roof along the entire length of the arena. The jet nozzles and the exhaust towers were also switched off to avoid large eddies. In consequence, the air supply was reduced to 115,000 m³ h⁻¹, corresponding to an air exchange rate of 0.85 h⁻¹.

*Defining participants with increased aerosol exposure.* The critical infectious dose and duration of stay threshold for an aerosol exposure-related infection is not yet known. Several studies have addressed this issue, but all have limitations, because many characteristics of SARS-CoV-2 (e.g. minimal infectious dose, virus concentrations in aerosols, etc) are not yet known[31–33]. We therefore used a pragmatic approach: a singing person emits about 1000 aerosol particles corresponding to $7.53 \times 10^{-8}$ ml of aerosols per second in our model[30]. In contrast, a resting person emits only a hundredth of particles[30]. We assumed that the viral load of the aerosols is equal to sputum with $10^9$ RNA copies/ml. Infectious visitors are most likely to be pre-symptomatic, or at the beginning of the symptomatic period, and the viral load is known to peak at this time point[34,35]. An infectious spectator thus emits ~$4 \times 10^5$ virus copies (=$7.53 \times 10^{-8}$ ml s⁻¹ × $10^9$ virus copies/ml × 5400 s) when singing during a 90 min concert and $4 \times 10^3$ virus copies (=$4 \times 10^5/100$) when resting. A threshold of 1% of the emission therefore corresponds to an exposure of 40 to 4000 virus particles per concert, which is the magnitude at which many assume the minimal infectious dose of Sars-CoV-2[36,37].

**Epidemiological simulation**
*Natural history model.* The model was developed as an extended susceptible-exposed-infectious-recovered (SEIR) model (Supplementary Fig. 9) using R (version 4.0.2). Susceptible individuals are moving from the state of being "exposed", to "infectious pre-symptomatic" and "infectious" at a rate as indicated in Supplementary Table 7. The disease of infectious individuals can progress stepwise to the more severe stages "hospitalized", "ICU admission", "death" or "recovery", with an age-dependent probability (Supplementary Table 8). A fraction of exposed people remain asymptomatic and have highly reduced infectiousness after the latent phase. The differentiation between asymptomatic, mild, and severe cases is independent of the compartments of the model and refers to the state in which the disease progresses. In our model, asymptomatic people, as well as mild and severe cases, undergo the different phases of disease, which are represented by the different compartments in Supplementary Fig. 9. This is necessary because the state of disease (susceptible, latent, pre-symptomatic, symptomatic, resistant) has effects on the infectiousness, and thus also applies to people with very unclear, ambiguous symptoms who would commonly be referred to as asymptomatic. Age-specific hospitalization rates were obtained from the Federal State of Schleswig Holstein[38]. Only severe cases are hospitalized in our model. Age-specific mortality was fitted to the respective rates for Germany[39]. We also assumed the same 7% contact risk of infection for all aerosol exposed individuals and direct contacts.

We simulated, in 1000 runs, the combined contacts identified via contact tracing and aerosol distribution for all three scenarios (Supplementary Table 4) with a calibrated baseline incidence of 10, 50, and 100 per 100,000 inhabitants/ 7 days, for an overall number of 100,000 and 200,000 event participants per 30 days (corresponding to events with 3300 and 6700 participants per day, the latter number corresponding to the pre-pandemic state). We also compared scenarios using masks in the MGE vs. no masks. For the impact on population level, a 30 day period was studied for the outcomes. This implies that outcomes resulting from the events were counted effectively over a shorter time period, as some late secondary and tertiary infections, as well as their late outcomes, might not develop within the studied time window.

*Contact network.* Model assumptions regarding daily contacts in the population were taken from the European POLYMOD contact study[21]. The age-specific contact rates were applied to the population of the city of Leipzig (Supplementary Table 8). We considered three types of contact settings for the model: household, school/work, and other (including the contact categories transport, leisure, and other from the POLYMOD study[21] (Supplementary Table 9). We assume that there is exactly one home place (household) for each person, and that there is exactly one place at day-care or school for each person aged 0–19 years. Each school or daycare class consists of 20 children (reflecting the average class size in Saxony), as well as one teacher aged 20–64 years old. For each person between 20 and 69 years of age (apart from teachers), there is one workplace. We included four different workplace sizes according to the categories reported by the Federal Statistical Office in Germany. The selection of a person's working place is random, but depends on age (based on additional analyses of POLYMOD data). We only considered contacts longer than 15 min for the numbers of contacts for each person, which is in line with the high-risk contact definition from the RKI. As a kind of fourth network, the persons attending an event are drawn randomly every day from the entire population. Only people who tested positive were excluded, as well as persons below 15 years of age. The numbers of both proximity and aerosol contacts within these events were provided by the experimental concert (Supplementary Table 4).

*Epidemic control measures.* Our model is based on the national guidelines for Germany regarding the SARS-CoV-2 testing strategy: testing due to symptoms or as someone who has been in contact with a known case. We assume perfect tests with 100% sensitivity and specificity. There are two ways to detect infected individuals in the model, either due to symptoms or due to contact tracing. Someone with a severe case of COVID-19 is assumed to be tested, detected, and isolated within one day of symptom onset, with an overall probability of 90%. Cases with mild symptoms are assumed to be tested within two days after symptom onset with an overall probability for testing of 50%. Asymptomatic individuals, who never develop symptoms, can only be detected by contact tracing. After a person has tested positive, the household members of that individual are requested to be tested within one day, and stay in quarantine for 14 days. We did not account for non-compliance during self-quarantine. We assumed a detection rate of 100% for all household members, thus adding new branches for contact tracing. In contrast, the testing rate is assumed to be 80% within the school or work network, with a delay of two days for the test results. In the third contact network "other", we assumed that only 50% of contacts were identified and tested, with a delay of four days. Infected persons are detected through contact tracing, if the individual passed the latent phase, including pre-symptomatic and completely asymptomatic persons. Contract tracing is triggered by the new detection of any infected person, and is independent if the detection itself was due to symptoms or past contact tracing. Recursive contact tracing can therefore result in the detection of whole infection chains. Any detected person is isolated for 14 days (including hospital stay if necessary). In this model, secondary contacts were not quarantined pre-emptively, which is in line with the national test strategy of Germany, but could be identified once the primary contact became a confirmed case.

*Reproduction number.* The POLYMOD contact matrix corresponds to the pre-pandemic state. We calibrated the average per contact transmission probability to obtain a reproduction number for the epidemic of about 3 (while assuming the susceptible fraction at 100%—i.e. conditions at the beginning of the epidemic) and subsequently reduced the contacts in all settings uniformly by 50% and applied the epidemic control measures, finally arriving at an R around 1. As testing is also included in the model, the model also provides an age-specific proportion of detected and undetected cases. With the intended incidence and an age-specific rate of detected and undetected cases, each run starts with a 14 day burn-in phase followed by the 30 day evaluation period. The POLYMOD contact matrix ignores the potential transmission of respiratory pathogens due to aerosols, which increases the probability of acquiring infection in a contact for a given reproduction number.

*Demographic background.* The study was conducted in Leipzig, which is a large city in the Federal State of Saxony in eastern Germany, with about 601,083 inhabitants[40]. The detailed demographics for the model are based upon Leipzig[40]. Since the duration of the simulated epidemic was less than a year, we did not consider changes in the population due to births, deaths, migration, or aging.

**Statistical analysis.** The mean and the standard deviation around the mean, the range, and interquartile range (IQR) were calculated for statistical analyses, as well as 95% confidence intervals.

**Reporting summary.** Further information on research design is available in the Nature Research Reporting Summary linked to this article.

## Data availability

Most of the quantitative results are provided in the supplemental tables. The raw contact data generated in this study have been deposited in the Zenodo database under accession code https://doi.org/10.5281/zenodo.5137667[41].

## Code availability

The codes for the contact tracer analysis and simulation model have been deposited in the Zenodo database under accession code https://doi.org/10.5281/zenodo.4647830 and https://doi.org/10.5281/zenodo.4770064[42,43].

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

## Acknowledgements

First, we especially thank Karsten Guenther, Managing Director of SC DHfK Handball, Leipzig, and Philipp Franke and Matthias Koelmel (Directors of the Quarterback Immobilien Arena), who provided the inspiration for setting up this study and supported the experiment, provided the arena and helped to recruit participants. We thank the musician Tim Bendzko who performed the concert. We also thank the governments of the Federal States of Saxony and Saxony-Anhalt who agreed to fund the project and the required tests. We thank all the helpers in the project, who prepared and organized the full day event. Finally, we would like to thank all participants of the experiment who donated their time for this project.

## Author contributions

S.M.: conceptualization, methodology, formal analysis, investigation, resources, writing—original draft, writing—review and editing, supervision, project administration, funding acquisition. C.G.: conceptualization, methodology, formal analysis, investigation, resources, writing—original draft, writing—review and editing, supervision, visualization. J.H.: methodology, software, formal analysis, investigation, resources, data curation, visualization, writing—original draft. M.P.: conceptualization, methodology, formal analysis, investigation, resources, data curation, writing—original draft. S.L.: investigation, resources, writing—original draft. B.K.: investigation, resources, writing—original draft. A.I.: conceptualization, resources. O.P.: software, formal analysis, data curation. M.G.: investigation, supervision, funding acquisition. F.Z.: CRD-conceptualization, methodology, investigation, formal analysis, writing - review and editing. R.M.: conceptualization, methodology, investigation, writing - review and editing, supervision, project administration.

## Funding

## Competing interests

The authors declare no competing interests.

## Ethics statement

The Ethics Committee of the Martin Luther University (Halle, Germany) approved the data collection for the RESTART-19 study (reference number 2020-095). The responsible authorities of the Federal State of Saxony (Saechsisches Staatsministerium fuer Soziales und gesellschaftlichen Zusammenhalt) provided special permission for the mass event. The Public Health Authority of the city of Leipzig approved the hygiene and safety practice for the event date.
