## [Peer Review File · Nature Communications]

REVIEWER COMMENTS

Reviewer #1 (Remarks to the Author):

Review of 'The risk of indoor sports and culture events for the transmission of COVID-19'

This paper describes a study whose goal was to quantify the risks that mass gathering events (MGE) might play in the transmission of SARS-CoV-2. The authors conducted an in-person gathering in which participants wore badges that could detect close proximity to one another. Volunteers who participated in this in-person gathering were asked to attend a rock concert under three different scenarios, intended to capture different possible levels of strictness. The results of this in-person study were combined with a simulation study that modeled airflow in the venue under two different scenarios; and an epidemiological model that simulated disease transmission dynamics. The goal was to estimate how many new infections would result from holding a MGE under each scenario.

In the big picture, this is an ambitious study that uses an interesting combination of methods to study a timely problem. However, I had several concerns about the current version of the paper. Most of these concerns focus on study design; I describe them in greater detail below.

* This design only considers transmission that would take place at the MGE itself. But if an MGE were held, it seems likely that this would also cause many other opportunities for transmission -- for example, transit to the concert venue (in cars or trains), restaurants or bars before/after the concert (if these are open), etc etc. This seems worth mentioning or discussing in the paper.

* The study was very focused on a specific arena, but the conclusions seem to be much broader. To support this, it would be useful if the authors explained how / why they expect these results to generalize to other settings that might host MGE. How typical is the shape and airflow in the Leipzig arena? How common are the ventilation systems in the arena? People's contact patterns entering and exiting Leipzig arena must be affected by the physical layout of the space -- is this layout common? These are important because the results of this study could potentially be used to justify holding MGE in different settings. So I suggest either adding an argument about how/why these findings generalize, or adding cautionary language about the conclusions of this study being specifically based on the Leipzig Arena.

* The participants were presumably aware that everyone had tested negative in order to participate in the study. It seems possible that this would affect how people behaved during the simulated concert. I would have been interested to see this concern discussed -- either it is a limitation of the study design, or perhaps there is a reason to think that participants' behavior was not different as a result of everyone's negative tests

* Similarly, I understand that the researchers designed three different 'hygiene concept' scenarios. How well did participants adhere to these scenarios? Is there a concern that participants followed these instructions more (or less) accurately than people who had not volunteered to participate in a scientific study would?

* From the SI, I see that the number of participants was much lower than the original target. The authors helpfully explain the factors that contributed to this lowered recruitment, but I was left wondering if having so many fewer participants was not a problem for the design. Do people's numbers of contacts and their behavior in the arena not change when the arena is pretty empty, as opposed to half full?

* Around line 113 - Related, it seems a little strange to parameterize these results in terms of the number of acquired infections; wouldn't these numbers be expected to scale with the size of the event? I would think that some kind of per-capita outcome would be more appropriate. Or, if that is not correct, I would have found it helpful if the authors explained why.

* I was a little surprised that this study passed ethical review, since there was obviously at least some risk to the participants. I see that this is discussed a little in the supporting materials - it might be worth mentioning the ethical review procedures in the main text, too.

* Throughout the paper, the writing could be cleaned up and edited for language; several passages are a little hard to read. I started making some specific notes but eventually gave up; here are a few minor details that the authors might consider:

* line 28 - 'total number of contacts per visitor' (typo)

* line 30 - '10fold' is a typo, I think

* line 31 - I find the term 'applied hygiene concept' to be a little confusing

* line 105 - I think this sentence would benefit from actually citing some serological studies

Reviewer #2 (Remarks to the Author):

In this manuscript, authors investigate the feasibility of reopening mass gathering events such as indoor sports and culture events by estimating transmission risk of SARS-COV-2 by conducting an experimental indoor mass gathering event combined with computational fluid dynamics simulation to estimate the aerosol transmission as well as an agent-based mathematical model to estimate the expected number of infections arising from such events. They compare their results of moderate and strong restrictions for indoor social distancing with no restrictions. Moreover, they combine their scenarios under two different ventilation scenarios. Authors found that under an effective ventilation system, indoor mass gathering events can be conducted safely with suitable measures such as compulsory N95 mask-wearing and socially distant seating arrangements. It is a well-conducted study and provides important insight for policy decision-makers. Please find my comments about the manuscript as follows:

1. To calculate the probability that an infected individual may attend an event, authors consider positivity rates (10,50, or 100 positive tested cases/100,000 population per week). However, the positivity rate may not be a true representation of the true number of cases due to under-reporting. Specifically, asymptomatic as well as pre-symptomatic individuals would not be well accounted for. However, policymakers will have to rely on the positivity rates to make policies. So, it makes sense to use them, but these numbers should be adjusted to account for under-reporting while calculating the risk of an infected individual attending an event. For example, an under-reporting of 10% would mean that a positivity rate of 10/100,000 should correspond to 11/100,000 cases for the model.

2. I don't see enough detail in the manuscript to calculate the expected number of infectious persons (7.8,37.8,75) attending an event? Is this calculated for each event? If there are N number of events each month, what would be the cumulative impact on the COVID-19 transmission in the community?

3. Calculating the number of infected individuals who may attend an event should be based on prevalence, rather than incidence? Why and how was incidence used? If weekly incidence was used as a proxy for infections currently in the community, it makes more sense to calculate it as the sum of daily incidence

4. If a guideline that promotes mask use rather than compulsory use is followed, it may provide a scenario where the effect of mask compliance can be assessed. Using results from the questionnaire (89% felt wearing an N95-mask is unproblematic), it is possible to run the model and compare it to understand the importance of compliance.

5. In Table 2: Incidence 10, S1-3 for Ventilation version 1, why is the mask the same or worse than no mask?

6. Asymptomatic infections (compartment) is not depicted in the model diagram (Extended Data Figure 5).

Paper ID: NCOMMS-21-00386-T

Paper title: The Risk of Indoor Sports and Culture Events for the Transmission of COVID-19

Comments:

In general, this paper contains three parts and presents: 1. the measurement results of the number of contacts during a MGE in the Leipzig Arena by conducting a pop concert under experimental conditions and providing all participants with a contact tracing device (CTD) under three different hygiene concepts; 2, the modelling results of indoor air flows, aerosol distribution and exposure of healthy subjects in the Leipzig Arena under two ventilation conditions; 3. the estimated results of the additional effect of indoor MGE on the overall burden of infections on a specific region using an individual based model.

These results may provide some new data on this hot topic – Transmission of COVID-19, especially the measurement results (number of contacts, acceptance of hygiene concepts). It can be seen that the authors paid great effort to design and conduct the field experiment. However, from the perspective of writing and presentation, the paper is not well written and not easy to follow in some sections. There are quite a few major issues that I would like to see addressed prior to the publication of this manuscript in this high quality journal, some of which relate to methodological aspects of the paper that affect the robustness of the data interpretation, which are outlined below:

1. It is hard to find the clear internal links between the three parts in the paper. It is not clearly sure whether the authors used the experiment results from part 1 in the part 2 modelling and part 3 estimating, as well as used the modelling results from part 2 in the part 3 estimating. More explanation may need.
2. For the Extended Data Table 2: A. The meaning of the data in Extended Data Table 2 is not clear. B. Are they the overall averages of the 1212 participants' number of contacts? C. For my understanding, the "Total value" should be the sum value of "Entry", "1st half", "half time", "2nd half" and "Exit". But it is unlikely based on the data in the table. For example: the total value is 63.9 in the first row, but the sum value is 84. The total value is 36.4 in the second row, but the sum value is 51.1, and so on. D. The calculation method of the cumulative data may need more explanation/clarification.
3. It may be useful if the authors can provide more detail number of contacts by different age group and gender.
4. For the section of "Simulation of aerosol exposure": 1. For my understanding, there are three main factors which control indoor aerosol distribution and exposure of healthy subjects, namely: air exchange rate (AER), mode of the airstream (or indoor air flow pattern (AFP)) and locations of the infectious person. The mode of the airstream of VV2 (air from ground to roof) should be better than that of VV1 (from roof to ground). However, this paper just gave an overall results, did not analyse the effect of each factor (AER, AFP), respectively. 2. The location arrangement of the 24 infectious persons may significant effect the results of the exposure of healthy subjects. 3. There is no information about how the authors validate their CFD model.

5. For the Table 2: From this table, we can see that the use of masks can decrease the positive cases than that without use of masks. However, we can also see that for MGE with 200000 participants, there is a 13.3%, 11.3%, and 7.7% increase of positive cases attributed to MGE for scenarios 1, 2 and 3 without the use of masks in VV1, which also increases to 13.3%, 11.6%, and 9.2% with masks at the same ventilation (VV1) and the same incidence setting (10 per 100000 per week). Based on these two group data, it seems that the use of masks may make more positive cases than that without use of masks. Is there any explanation for these contradiction results?
6. For the section of “Effect of mass gathering events on SARS-CoV-2 positive cases in the population”: I am sorry that I could not provide my comment as this part (Epidemiological simulation) is not in my research areas.
7. Page 19, Line 343: It is not sure whether the ICDWpro quad 164643 tags can measure the distances among more than two participants, simultaneously. It means whether a ICDWpro quad 164643 tag can measure more than one targets at the same time. If not, how to deal with the situation that there are more than two participants in the distance of <1.5m at the same time period? Such as, three participants (A, B, C) stand together within the distance of <1.5m.
8. Page 21, Line 392: I did not find the information “emission rate” from the Ref 23.
9. Page 21, Line 393, There is not “Hartmann et al. and Stadnytskyi et al.” in the References list. I also did not find the information “physical density of respiratory air” from the Ref 24 and Ref 25.
10. Page 22, Line 426: As above, I did not find the information about aerosol emission rate from the Ref 23.
11. Page 22, Line 426: I did not find the information about aerosol emission rate from the Ref 16.
12. Page 27: From the Author contributions section, it is hard to see who contributed the part 2 (the modelling of indoor air flows, aerosol distribution and exposure of healthy subjects). If none of the authors’ research area is in aerosol science and aerosol modelling, it may be better that Frank Zimmermann will be as one of the authors in this paper.
13. For the Extended Data Table 7: The parameter “with n” needs clarify. What its unit is?

Response to reviewers' comments to the manuscript: "The risk of indoor sports and culture events for the transmission of COVID-19"

Please note that our responses are interspaced and in bold letter

Comments to the reviewers

Reviewer #1 (Remarks to the Author):

This paper describes a study whose goal was to quantify the risks that mass gathering events (MGE) might play in the transmission of SARS-CoV-2. The authors conducted an in-person gathering in which participants wore badges that could detect close proximity to one another. Volunteers who participated in this in-person gathering were asked to attend a rock concert under three different scenarios, intended to capture different possible levels of strictness. The results of this in-person study were combined with a simulation study that modeled airflow in the venue under two different scenarios; and an epidemiological model that simulated disease transmission dynamics. The goal was to estimate how many new infections would result from holding a MGE under each scenario.

In the big picture, this is an ambitious study that uses an interesting combination of methods to study a timely problem. However, I had several concerns about the current version of the paper. Most of these concerns focus on study design; I describe them in greater detail below.

* This design only considers transmission that would take place at the MGE itself. But if an MGE were held, it seems likely that this would also cause many other opportunities for transmission -- for example, transit to the concert venue (in cars or trains), restaurants or bars before/after the concert (if these are open), etc etc. This seems worth mentioning or discussing in the paper.

>> We agree and mention this now in the limitations (line 227):

"Third, we did not analyse other opportunities for contacts which could be linked to the MGE. For example, additional contacts could take place during transit, or if participants of the MGE go to bars or similar venues after the concert. We assumed that all the other settings would have their own hygiene practices, for example, not allowing overcrowding. In a practical sense, this can be difficult for large events, but with additional efforts the excess risk can be minimized."

* The study was very focused on a specific arena, but the conclusions seem to be much broader. To support this, it would be useful if the authors explained how / why they expect these results to generalize to other settings that might host MGE. How typical is the shape and airflow in the Leipzig arena? How common are the ventilation systems in the arena? People's contact patterns entering and exiting Leipzig arena must be affected by the physical layout of the space -- is this layout common? These are important because the results of this study could potentially be used to justify holding MGE in different settings. So I suggest either adding an argument about how/why these findings generalize, or adding cautionary language about the conclusions of this study being specifically based on the Leipzig Arena.

>> We thank the reviewer for this comment. Our aim was to generate a more general knowledge, while we had to limit ourselves to a specific example. The setting of the experiment followed the typical pattern of the phases entry, halves, break, exit. Here, we think that the general strategy of reducing contacts can be also transferred to other settings, even if the physical layout is different. The experiment venue belongs to the typical class. We added the corresponding information in the manuscript (line 347):

“The Quarterback Immobilien Arena is an event location in the city of Leipzig and is one of the 10 most frequented live entertainment venues in Germany (<https://www.stadionwelt.de/plus/arena-ranking-besucher>). The type and layout of the arena (multipurpose hall) is common in the industry, and further examples can be found in Stuttgart (Porsche Arena), Berlin (Max Schmeling Arena) and Nuremberg (Arena Nuremberg Insurance).”

There might be more variation with respect to the ventilation systems. We added this information in the limitations’ section (line 221): “The ASHRAE Standard 62.1-2013 recommends a minimum ventilation rate of 3.8 l/s per person in the spectator area. Due to an air supply of 198 000 m³/h the Leipzig arena supplies 6.7 l/s per person (max. 8200 seating persons) or 4.5 l/s (max. 12,300 standing persons), and exceeds the ASHRAE recommendation by approximately 1.75 times or 1.2 times, respectively. Since ventilation is crucial for the risk associated with MGEs, it is important that further studies focus on this aspect.”

* The participants were presumably aware that everyone had tested negative in order to participate in the study. It seems possible that this would affect how people behaved during the simulated concert. I would have been interested to see this concern discussed -- either it is a limitation of the study design, or perhaps there is a reason to think that participants' behavior was not different as a result of everyone's negative tests.

>> We agree that the knowledge about the negative test could lead to a more “liberal” behavior. However, our experience was that the participants behaved in a very responsible way. We also pointed out that hygiene stewards should be part of a hygiene concept for such events. We added a comment on this aspect in the discussion (line 207): “Hygiene stewards rarely had to intervene in our experiment. This might be a consequence of the highly disciplined participants in our study, but also indicates that knowledge about being tested negatively did not lead to the breach of specific distancing rules in the various phases of the experiment.”

* Similarly, I understand that the researchers designed three different 'hygiene concept' scenarios. How well did participants adhere to these scenarios? Is there a concern that participants followed these instructions more (or less) accurately than people who had not volunteered to participate in a scientific study would?

>> To ensure that people adhered strictly to the different hygiene concepts, we employed 40 hygiene stewards throughout the arena that had the authority of dismissing participants from the event in case of non-adherence. We stated that such rule should be part of a hygiene concept. In fact, our impression was that the participants adhered well to the various scenarios. In a real case, there would be only one scenario, so the rules would be more easily to follow. We added a comment among the limitations (line 235): “Fourth, lack of adherence to hygiene practices is a potential danger, but reinforcing the hygiene practice should be a requirement for MGEs.”

* From the SI, I see that the number of participants was much lower than the original target. The authors helpfully explain the factors that contributed to this lowered recruitment, but I was left wondering if having so many fewer participants was not a problem for the design. Do people's numbers of contacts and their behavior in the arena not change when the arena is pretty empty, as opposed to half full?

>> Because all participants had to register before, we knew how many people to expect in the event and prepared the arena accordingly well in advance. We divided the arena into four seating quadrants, closed two quadrants in scenario 1 completely, and closed half of the upper ranks of the open quadrants to create a realistic density. We also closed a respective number of entries, catering stalls, and bathrooms and proceeded accordingly in scenarios 2 and 3 to create a realistic setting. We added the following to the Materials and Methods (line 322): “Because fewer individuals participated in the event than initially planned, we prepared the arena to create a setting of realistic density: we closed seating ranks, catering stalls, bathrooms, and entries as required by each hygiene practice.”

* Around line 113 - Related, it seems a little strange to parameterize these results in terms of the number of acquired infections; wouldn't these numbers be expected to scale with the size of the event? I would think that some kind of per-capita outcome would be more appropriate. Or, if that is not correct, I would have found it helpful if the authors explained why.

>> The reported numbers reflect the total number of excess reported cases for the 200 000 persons taking part in the MGE during a month. In this way, it is related to the number of participants. We acknowledge that our description was somewhat unclear as this are not only persons who directly acquired infections in the event, but also secondary cases. At the same time, the numbers refer only to cases recorded in the system.

The text in the manuscript was changed to (line 150): “The resulting additional average numbers of persons who would become infected and would be detected (excess cases) ranges from 5.1 under the strictest hygiene practice and best ventilation (Scenario 3, VV1) to 22.0 with no hygiene practice and non-optimal ventilation (Scenario 1, VV2) in the low incidence scenario (10 per 100,000 per week) and with spectators wearing masks (Table 1).”

* I was a little surprised that this study passed ethical review, since there was obviously at least some risk to the participants. I see that this is discussed a little in the supporting materials - it might be worth mentioning the ethical review procedures in the main text, too.

>> Thank you for pointing out the additional need for details regarding ethical approval. We provided more details in our ethics statement (line 603): “The Ethics Committee of the Martin Luther University (Halle, Germany) approved the data collection for the RESTART-19 study. The responsible authorities of the Federal State of Saxony (Saechsisches Staatsministerium fuer Soziales und gesellschaftlichen Zusammenhalt) provided special permission for the mass event. The Public Health Authority of the city of Leipzig approved the hygiene and safety practice for the event date.”

* Throughout the paper, the writing could be cleaned up and edited for language; several passages are a little hard to read. I started making some specific notes but eventually gave up; here are a few minor details that the authors might consider:

* line 28 - 'total number of contacts per visitor' (typo)

* line 30 - '10fold' is a typo, I think

* line 31 - I find the term 'applied hygiene concept' to be a little confusing

>> We corrected the typos mentioned above.

* line 105 - I think this sentence would benefit from actually citing some serological studies

>> Please find the corresponding references in the text and below:

16. Harvey, R. A. et al. Association of SARS-CoV-2 Seropositive Antibody Test with Risk of Future Infection. *JAMA Intern. Med.* 20850, 1–7 (2021).
17. Robert-Koch-Institut (RKI) (2021). Serologische Untersuchungen von Blutspenden auf Antikörper gegen SARS-CoV-2 (SeBluCo-Studie). Available at: https://www.rki.de/DE/Content/InfAZ/N/Neuartiges_Coronavirus/Projekte_RKI/SeBluCo_Zwischenbericht.html (Accessed: 15 May 2021).
18. Fischer, B., Knabbe, C. & Vollmer, T. SARS-CoV-2 IgG seroprevalence in blood donors located in three different federal states, Germany, March to June 2020. *Eurosurveillance* 25, 1–4 (2020).
19. Bajema, K. L. et al. Estimated SARS-CoV-2 Seroprevalence in the US as of September 2020. *JAMA Intern. Med.* 30329, 1–11 (2020).
20. Slot, E. et al. Low SARS-CoV-2 seroprevalence in blood donors in the early COVID-19 epidemic in the Netherlands. *Nat. Commun.* 11, 1–7 (2020).

>> Thank you for pointing out these mistakes. We have corrected them and asked for professional language proofreading. We think the manuscript has benefited much from that.

Reviewer #2 (Remarks to the Author):

In this manuscript, authors investigate the feasibility of reopening mass gathering events such as indoor sports and culture events by estimating transmission risk of SARS-COV-2 by conducting an experimental indoor mass gathering event combined with computational fluid dynamics simulation to estimate the aerosol transmission as well as an agent-based mathematical model to estimate the expected number of infections arising from such events. They compare their results of moderate and strong restrictions for indoor social distancing with no restrictions. Moreover, they combine their scenarios under two different ventilation scenarios. Authors found that under an effective ventilation system, indoor mass gathering events can be conducted safely with suitable measures such as compulsory N95 mask-wearing and socially distant seating arrangements. It is a well-conducted study and provides important insight for policy decision-makers. Please find my comments about the manuscript as follows:

1. To calculate the probability that an infected individual may attend an event, authors consider positivity rates (10, 50, or 100 positive tested cases/100,000 population per week). However, the positivity rate may not be a true representation of the true number of cases due to under-reporting. Specifically, asymptomatic as well as pre-symptomatic individuals would not be well accounted for. However, policymakers will have to rely on the positivity rates to make policies. So, it makes sense to use them, but these numbers should be adjusted to account for under-reporting while calculating the risk of an infected individual attending an event. For example, an under-reporting of 10% would mean that a positivity rate of 10/100,000 should correspond to 11/100,000 cases for the model.

>> We agree and thank for the opportunity to explain it better. In fact, our incidence refers to the observed incidence in Germany. We allowed for a fraction of undetected cases, and also that some cases are not detected at the time point of the event yet,

although they might be detected later. Our model referred to the situation of an effective control of the epidemic, therefore the corresponding fraction of undetected cases was calibrated at about 50%. We added this information in the manuscript (line 135): “For certain predefined incidences (based on diagnosed cases), we simulated the number of infectious persons attending the MGE. Per definition, these persons were not diagnosed yet, and displayed no symptoms at the time of the MGE – otherwise they would not attend the event. Similarly, quarantined persons are excluded, and (asymptomatic) cases identified by contact tracing could not attend the event. Conversely, undetected asymptomatic, pre-symptomatic and some fraction of mildly symptomatic persons could attend the event. Some of these persons would remain undiagnosed and contribute to the dark figure of cases. The testing strategy and control measures resulted in a controlled epidemic, in which about 50% of the infected were detected during the course of infection. The number of infectious MGE participants was obtained from the agent-based model accounting for these aspects.”

The number of persons infectious during the event was estimated for all infectious cases independent of whether they were detected. In fact, persons who were positively tested would be prohibited from attending the event.

2. I don't see enough detail in the manuscript to calculate the expected number of infectious persons (7.8,37.8,75) attending an event? Is this calculated for each event? If there are N number of events each month, what would be the cumulative impact on the COVID-19 transmission in the community?

We thank the reviewer for this question. The cited numbers are those aggregated for events which in total have 200 000 participants per month – the same is also true for resulting cases. However, the expected infectious cases are “true” in the sense that they include unreported incidence and in contrast, resulting excess cases are reported cases to be comparable with the observed incidence. We clarified now that the expected numbers of infectious persons are the output of the model and cannot be directly calculated from the observed incidence. We simulated a state of controlled epidemics, with contact reduction, but also testing of mild symptomatic cases and well working contact tracing. We rewrote the corresponding section to make it clearer (line 145): “Depending on the incidence (i.e. 10, 50 or 100 positive tested cases per/100,000 population per week), on average 7.8, 37.8 or 75 infectious persons might attend any event, assuming the total number of persons taking part in MGE is 200,000 per month (corresponding to the number of participants in MGE in Leipzig in pre-pandemic times, Extended Data Figure 3).”

3. Calculating the number of infected individuals who may attend an event should be based on prevalence, rather than incidence? Why and how was incidence used? If weekly incidence was used as a proxy for infections currently in the community, it makes more sense to calculate it as the sum of daily incidence.

>> As explained above, the number of infectious individuals who attend the MGE is provided by the model in such way that they correspond to the observed incidence. Clearly, these are prevalent cases at the time of the event, but by definition they are at this point undetected yet. We hope that the clarifications provided in response to the previous points will also address this point.

4. If a guideline that promotes mask use rather than compulsory use is followed, it may provide a scenario where the effect of mask compliance can be assessed. Using results from

the questionnaire (89% felt wearing an N95-mask is unproblematic), it is possible to run the model and compare it to understand the importance of compliance.

>> We added a comment on this aspect in the discussion (line 196): “We compared MGEs with and without mask wearing, and, since the effects are proportional within an MGE, the effect of wearing masks by any percentage of the participants can be directly estimated.”

5. In Table 2: Incidence 10, S1-3 for Ventilation version 1, why is the mask the same or worse than no mask?

>> In the scenario with an incidence of 10 per 100000 random effects have such a strong impact on the disease that the impact of masks is negligible.

6. Asymptomatic infections (compartment) is not depicted in the model diagram (Extended Data Figure 5).

>> We added an explanation in the Materials and Methods section (line 522). “The differentiation between asymptomatic, mild and severe cases is independent of the compartments of the model and refers to the state in which the disease progresses. In our model, asymptomatic people, as well as mild and severe cases, undergo the different phases of disease, which are represented by the different compartments in Extended Data Figure 5. This is necessary because the state of disease (susceptible, latent, pre-symptomatic, symptomatic, resistant) has effects on the infectiousness, and thus also applies to people with very unclear, ambiguous symptoms who would commonly be referred to as asymptomatic.”

However, we modeled asymptomatic cases differently regarding testing which is already described in the Materials and Methods section. Additionally, only severe cases will need hospitalization in our model. We added (line 530) “Only severe cases are hospitalized in our model.”

Reviewer #3 (Remarks to the Author):

Comments:

In general, this paper contains three parts and presents: 1. the measurement results of the number of contacts during a MGE in the Leipzig Arena by conducting a pop concert under experimental conditions and providing all participants with a contact tracing device (CTD) under three different hygiene concepts; 2, the modelling results of indoor air flows, aerosol distribution and exposure of healthy subjects in the Leipzig Arena under two ventilation conditions; 3. the estimated results of the additional effect of indoor MGE on the overall burden of infections on a specific region using an individual based model. These results may provide some new data on this hot topic – Transmission of COVID-19, especially the measurement results (number of contacts, acceptance of hygiene concepts). It can be seen that the authors paid great effort to design and conduct the field experiment. However, from the perspective of writing and presentation, the paper is not well written and not easy to follow in some sections. There are quite a few major issues that I would like to see addressed prior to the publication of this manuscript in this high quality journal, some of which relate to methodological aspects of the paper that affect the robustness of the data interpretation, which are outlined below:

>> Thank you for your comments. Please see our response below. In addition, we also asked for professional proofreading. Our manuscript has benefited much from that.

1. It is hard to find the clear internal links between the three parts in the paper. It is not clearly sure whether the authors used the experiment results from part 1 in the part 2 modelling and part 3 estimating, as well as used the modelling results from part 2 in the part 3 estimating. More explanation may need.

>> We included an Extended Data Table 9, which describes the calculated number of contacts in an event per scenario and per ventilation system by direct contact measured using CTD and modelled aerosol contact. These numbers are used as number of contacts for each person in an event in the epidemiologic model. We refer to this table throughout the manuscript, i.e. in the following sentence (line 533): “We simulated, in 1000 runs, the combined contacts identified via contact tracing and aerosol distribution for all three scenarios (Extended Data Table 9) with a calibrated baseline incidence of 10, 50, and 100 per 100,000 inhabitants/7 days, for an overall number of 100,000 and 200,000 event participants per 30 days (corresponding to events with 3300 and 6700 participants per day, the latter number corresponding to the pre-pandemic state).”

Extended Data Table 9. Number of exposed individuals by direct contact ($\leq 1.5\text{m}$) and through aerosol exposure for each ventilation version (VV). SD = standard deviation.

Type of exposure	Mean number of exposed per each infected individual (\pm SD) in VV 1			Mean number of exposed per infected individuals (\pm SD) in VV 2		
	Scenario 1	Scenario 2	Scenario 3	Scenario 1	Scenario 2	Scenario 3
Direct contact	9.0 (± 3.5)	4.7 (± 1.9)	1.3 (± 0.9)	9.0 (± 3.5)	4.7 (± 1.9)	1.3 (± 0.9)
Aerosols	3.5 (± 2.9)	1.9 (± 1.5)	0.7 (± 1.0)	25.5 (± 27.8)	11.8 (± 13.5)	5.3 (± 6.4)
Total	12.5	6.6	2.0	34.5	16.5	6.6

2. For the Extended Data Table 2: A. The meaning of the data in Extended Data Table 2 is not clear. B. Are they the overall averages of the 1212 participants’ number of contacts? C. For my understanding, the “Total value” should be the sum value of “Entry”, “1st half”, “half time”, “2nd half” and “Exit”. But it is unlikely based on the data in the table. For example: the total value is 63.9 in the first row, but the sum value is 84. The total value is 36.4 in the second row, but the sum value is 51.1, and so on. D. The calculation method of the cumulative data may need more explanation/clarification.

>> A/B/D. Yes, they are the overall averages of the 1212 participant’s number of contacts. We have changed the table title (line 825) to “Mean number of contacts of all participants, measured using mobile contact tracing devices.” and the legend to include more information on the cumulative number of contacts.

C. The total number of mean contacts (first column) is the equivalent of the total number of cumulative contacts (last column). The sum value of “Entry”, “1st half”, “half time”, “2nd half” and “Exit” is not the same, because here all contacts are counted, independent of whether the same contact has occurred before, i.e. a contact in 1st half can occur in 2nd half and would thus be counted twice if the number of all settings were summed up, but is counted only once in the first column “Total”. We have changed the legend accordingly (see below).

Regarding A/B/C/D, the legend was improved to (line 827): “The mean number of total contacts longer than 10 seconds, 5 minutes and 15 minutes are shown for all participants, as well as contacts stratified by setting (entry, 1st half, half time, 2nd half, exit) within all scenarios (1, 2, and 3). Contacts can be counted more than once, depending on which setting they occur in, but are counted once in total. Cumulative data is provided by adding additional contacts within a setting to the previous number of contacts. SD = standard deviation, S = scenario.”

3. It may be useful if the authors can provide more detail number of contacts by different age group and gender.

>> We analyzed the data stratified by different age groups and gender and found no difference regarding the number of contacts. We added this information to the manuscript (line 87): “Overall, no effect of gender or age was observed regarding the number of contacts during the event (Extended Data Figure 1).” and included Extended Data Fig. 1.

4. For the section of “Simulation of aerosol exposure”: 1. For my understanding, there are three main factors which control indoor aerosol distribution and exposure of healthy subjects, namely: air exchange rate (AER), mode of the airstream (or indoor air flow pattern (AFP)) and locations of the infectious person. The mode of the airstream of VV2 (air from ground to roof) should be better than that of VV1 (from roof to ground). However, this paper just gave an overall results, did not analyse the effect of each factor (AER, AFP), respectively. 2. The location arrangement of the 24 infectious persons may significant effect the results of the exposure of healthy subjects. 3. There is no information about how the authors validate their CFD model.

>> We thank the reviewer for this comment. We respond to the three points in the following. We also added a large paragraph to “Simulation of aerosol exposure” in the manuscript (see below) and further addition information in the Material and Methods section.

1. We agree with the assessment of the main factors influencing the aerosol distribution which are the amount of fresh air supplied (expressed as air exchange rate (AER)), indoor airflow pattern (AFP, turbulent vs. laminar) and the location of the infectious persons. Other important factors are accumulation time (event time), volume of the room and number of people in a given room. These factors were constant throughout the simulations.

The aim of the study was to examine the effect of two different ventilation strategies (VV1, VV2) on the aerosol distribution. We modified both the airflow pattern (from a more turbulent state to a more laminar state) and the air supply. However, we did not investigate each factor separately. We added to the Materials and Methods (line 491): “In the first ventilation variant, we modelled the actual ventilation system of the arena. Here, air reaches the arena via the outlets under the lateral grandstands and the jet nozzles, as described above. Two towers are installed in each corner of the hall for air suction. The air supply was 198,000 m³/h, corresponding to an air exchange rate of 1.46 air changes per hour (ACH). In a second variant, we tried to modify the ventilation system in the arena to enable displacement flow by virtually installing two long exhaust pipes under the roof along the entire length of the arena. The jet nozzles and the exhaust towers were also switched off to avoid large eddies. In consequence, the air supply was reduced to 115,000 m³/h, corresponding to an air exchange rate of 0.85 h⁻¹.”

To the manuscript, we added (line 94): “Air supply was also issued under the seats of the grandstands through swirl diffusers, and below the mobile grandstands through ventilation grilles. The exhaust air was discharged in the corners of the arena by exhaust towers. Air exchange per hour (ACH) was 1.46 h⁻¹, with a make-up air of 50m³/h-person. The make-up air is defined as the amount of air provided to a person in a room in one hour. To avoid large eddies (Extended Data Figure 7), which generate the intensified spread of aerosols at face level, jet nozzles and exhaust towers were turned off in Ventilation Version 2 (VV2) and the exhaust towers were replaced by exhaust pipes located under the roof, resulting in an ACH of 0.85 h⁻¹. This solution was chosen because its implementation would be cost-efficient, and in the hope that a displacement flow in the direction of the roof would be created by buoyancy induced flow.

Unfortunately, the buoyancy induced flow was too weak due to the low occupancy and interfering air supply nearby the grandstands. Stationary eddies also emerged above the grandstands (Extended Data Figure 8).”

The third factor influencing exposure to infectious aerosols is the location of the infectious individuals. Of course, the location of these is very important. To demonstrate the heterogeneity we included Table 2 in the manuscript. In addition, we added Extended Data Figure 10 on isosurfaces of the aerosol distribution, where the reader can see the different effects for each location.

We added additional information on CFD simulation to the results part “Simulation of aerosol exposure” (see below) and modified the methods part of the different ventilation strategies in the section “Variant of Ventilation” in Materials and Methods (line 491, see above).

2. The location arrangement of the 24 infectious individuals was done in that way to represent all areas of the arena, including stalls, mobile and stable grandstands to simulate aerosol emission and exposure in the breathing air. Positions of the infectious persons were determined according to a pre-calculation for the analysis of the room air flow. Since the flow pattern is almost identical from the first row of stalls to the last, the infectious persons were placed in the front and in the back of the stalls in such a way that the different room air flows in the stalls and on the stands were recorded. In such way, important variation was considered.

3. For aerosol modeling, we used the commercial CFD software PHOENICS with the add-ons GENTRA (particle tracking) and FLAIR (aerosol distribution). The software has been successfully used for more than 30 years by Zimmermann and Becker GmbH (Consulting Engineers, who carried out aerosol modeling). The PHOENICS/ FLAIR/ GENTRA model was used in numerous other studies focusing on CFD simulation of aerosols. We have added references to the appropriate passages.

For the aerosol modeling, we performed pre-calculations for the analysis of the room airflow to determine the positions of the infectious persons. Furthermore, we have carried out pre-calculations in a simplified model for the analysis of aerosol movement. For aerosols with $D \leq 10 \mu\text{m}$ as well as for CO₂, these calculations showed a direct dependence of the trajectory on the room air flow already from 0.05 m/s air velocity. Additional information about the model validation is now provided in the material and method section “Computational Fluid Dynamics” (line 445): “All CFD simulations were conducted with the commercial CFD software PHOENICS (Version 2020, CHAM, London, United Kingdom). For aerosol distribution the PHOENICS add-on FLAIR (drift-flux modelling), and for particle tracking the add-on GENTRA were used (both CHAM, London, United Kingdom). The PHOENICS program was developed by Professor Brian Spalding, and has been successfully used for more than 30 years by

Zimmermann und Becker GmbH, Consulting Engineers (Germany) for the validation of technical planning, in a wide range of technical applications, and is validated for new applications. The PHOENICS/ FLAIR/ GENTRA model has been used in numerous other studies focusing on the CFD simulation of aerosols^{25–28}. The Quarterback Immobilien Arena was exactly transferred into a 3D model (1:1), including all built-on components, the complete ventilation system, grandstands and seats. Virtual spectators were seated within the arena to simulate the aerosol emission and exposure. Virtual spectators were placed in the arena to simulate aerosol emission and exposure in the breathing air. The position of the infectious persons was determined according to a pre-calculation for the analysis of the room air flow (Extended Data Figure 7). Further pre-calculations were carried out in a simplified model for the analysis of aerosol movement. These calculations showed that the trajectory was directly dependent on room air flow from 0.05 m/s air velocity for aerosols with $d \leq 10 \mu\text{m}$ as well as for CO₂. Since the flow pattern is almost identical from the first row of stalls to the last, infectious persons were placed in the front and in the back of the stalls in such a way that the different room air flows in the stalls and on the stands were recorded. Due to the size of the model and the required accuracy of the grid, very long computation times were expected.”

Regarding the aforementioned issues, we added the following to the manuscript (line 90): “In addition to the number of contacts measured by CTD, the aerosol distribution in the respiratory air of all 4000 virtual participants was simulated using a computational fluid dynamics (CFD) model, considering two different ventilation versions (VV). Ventilation Version 1 (VV1) represented the current ventilation system in the arena. Here, the inlet air is blown in laterally on the east- and west side by jet nozzles (Movie S1). Air supply was also issued under the seats of the grandstands through swirl diffusers, and below the mobile grandstands through ventilation grilles. The exhaust air was discharged in the corners of the arena by exhaust towers. Air exchange per hour (ACH) was 1.46 h⁻¹, with a make-up air of 50m³/h-person. The make-up air is defined as the amount of air provided to a person in a room in one hour. To avoid large eddies (Extended Data Figure 7), which generate the intensified spread of aerosols at face level, jet nozzles and exhaust towers were turned off in Ventilation Version 2 (VV2) and the exhaust towers were replaced by exhaust pipes located under the roof, resulting in an ACH of 0.85 h⁻¹. This solution was chosen because its implementation would be cost-efficient, and in the hope that a displacement flow in the direction of the roof would be created by buoyancy induced flow. Unfortunately, the buoyancy induced flow was too weak due to the low occupancy and interfering air supply nearby the grandstands. Stationary eddies also emerged above the grandstands (Extended Data Figure 8). In the VV1 ventilation strategy, 24 infectious persons placed in the arena resulted in 85 individuals exposed to infectious aerosols, and the number for ventilation strategy VV2 was substantially higher (612 persons, Table 2). The overall mean exposure increased from 3.54 (VV1) to 25.50 exposed individuals (VV2), also resulting in a seven-fold increase in total. The mean exposure in the stalls was 6.75 (VV1) compared to 24.25, resulting in an almost four-fold increase in exposed individuals. In the mobile grandstands, the mean exposure increased from 10.25 (VV1) to 59.75, resulting in a six-fold increase. In the solid grandstands, the mean exposure changed from 4.25 (VV1) to 69.0 (VV2). Here, the increase was highest (16-fold), compared to the other areas (Table 2, Extended Figure 9). Extended Data Figure 10 shows the aerosol concentration displayed as isosurfaces around the infectious individuals. The isosurfaces show the same transmission mechanism for both VV1 and VV2, irrespective of the position: direct aerosol flow from the mouth of the transmitting individual to the mouth of the recipient. Differences in the number of infected individuals between the two ventilation

variants can be explained by the lower air exchange per hour (ACH) of 0.85 h⁻¹, as well as less air movement (and therefore slower mixing of the air) in VV2.”

5. For the Table 2: From this table, we can see that the use of masks can decrease the positive cases than that without use of masks. However, we can also see that for MGE with 200000 participants, there is a 13.3%, 11.3%, and 7.7% increase of positive cases attributed to MGE for scenarios 1, 2 and 3 without the use of masks in VV1, which also increases to 13.3%, 11.6%, and 9.2% with masks at the same ventilation (VV1) and the same incidence setting (10 per 100000 per week). Based on these two group data, it seems that the use of masks may make more positive cases than that without use of masks. Is there any explanation for these contradiction results?

>> We added the following explanation as footnote into Table 3 (formerly Table 2): “At an incidence of 10 per 100,000 people, random effects have a strong impact on the number of additional cases due to MGEs, such that the variation supersedes the impact of masks. The mean across all 1000 runs is therefore affected by stochastic fluctuation”

6. For the section of “Effect of mass gathering events on SARS-CoV-2 positive cases in the population”: I am sorry that I could not provide my comment as this part (Epidemiological simulation) is not in my research areas.

7. Page 19, Line 343: It is not sure whether the ICDWpro quad 164643 tags can measure the distances among more than two participants, simultaneously. It means whether a ICDWpro quad 164643 tag can measure more than one targets at the same time. If not, how to deal with the situation that there are more than two participants in the distance of <1.5m at the same time period? Such as, three participants (A, B, C) stand together within the distance of <1.5m.

>> We thank for identifying this imprecise description. We added the following sentence:

“Distances were not measured constantly, but at specified intervals (around every 3 seconds). The tags coordinate their broadcast time to minimize interference (only distant tags broadcast at the same time). Therefore, very short encounters can be missed (<3 sec), but longer encounters are recorded.”

8. Page 21, Line 392: I did not find the information “emission rate” from the Ref 23.

>> Thank you very much for your helpful hints. We corrected them all and referred to the right literature in the following.

We changed the references:

- **Hartmann *et al.* TU Berlin. Emission rate and particle size of bioaerosols during breathing, speaking and coughing. DOI: <http://dx.doi.org/10.14279/depositonce-10331>**
- **Mürbe D *et al.* Erhöhung der Aerosolbildung beim professionellen Singen: https://audiologie-phoniatrie.charite.de/fileadmin/user_upload/microsites/m_cc16/audiologie/Allgemein/muerbe_etal_2020_aerosole-singen_v2.pdf**

9. Page 21, Line 393, There is not “Hartmann et al. and Stadnytskyi et al.” in the References list. I also did not find the information “physical density of respiratory air” from the Ref 24 and Ref 25.

>> For better understanding, we rewrote the section (line 469) and added a reference that includes the information: “The breathing air of all virtual spectators consists of an ideal gas and 20 litres CO₂ per hour. Infectious spectators also emit aerosols of different sizes (0.5, 5 and 10 µm) into the room while breathing. Emission rate and particle size were adjusted to that of singing people^{29,30}. The assumed physical density of respiratory air was 1.3 g/ml³¹.”

31. Stadnytskyi, V., Bax, C. E., Bax, A. & Anfinrud, P. The airborne lifetime of small speech droplets and their potential importance in SARS-CoV-2 transmission. Proc. Natl. Acad. Sci. U. S. A. 117, 11875–11877 (2020).

10. Page 22, Line 426: As above, I did not find the information about aerosol emission rate from the Ref 23.

>> We changed the reference for better understanding:

30. Mürbe, D., Fleischer, M., Lange, J., Rotheudt, H. & Kriegel, M. Aerosol emission is increased in professional singing. Preprint 1–10 (2020) doi:10.31219/osf.io/znjeh.

11. Page 22, Line 426: I did not find the information about aerosol emission rate from the Ref 16.

>> We changed the reference for better understanding:

30. Mürbe, D., Fleischer, M., Lange, J., Rotheudt, H. & Kriegel, M. Aerosol emission is increased in professional singing. Preprint 1–10 (2020) doi:10.31219/osf.io/znjeh.

12. Page 27: From the Author contributions section, it is hard to see who contributed the part 2 (the modelling of indoor air flows, aerosol distribution and exposure of healthy subjects). If none of the authors' research area is in aerosol science and aerosol modelling, it may be better that Frank Zimmermann will be as one of the authors in this paper.

>> In the current version of this manuscript, Mr. Zimmermann who conducted the modelling, rewrote the corresponding part and is now listed as a co-author.

13. For the Extended Data Table 7: The parameter “with n” needs clarify. What its unit is?

>> Thank you very much for this hint! We deleted this row, as it is redundant, because “n” is already defined in the table.

REVIEWER COMMENTS

Reviewer #1 (Remarks to the Author):

The authors have addressed the concerns that I raised in my review.

Reviewer #2 (Remarks to the Author):

Thanks for addressing my comments. I am happy with the responses.

Reviewer #3 (Remarks to the Author):

Comments:

Thanks the authors' great effort to provide helpful responses, add some new meaningful tables and figures, as well as improve the quality of this manuscript. However, there are still some issues as following that I would like to see addressed prior to the publication of this manuscript in this high quality journal:

Line 85-84: This sentence may be not safe for the case: >10sec in Scenarios 2 and 3. Because the values of "half time" (24.7, 13.3) are about 100% (83%, 118%) higher than the values of "Entry" (13.5, 6.1).

Line 110: It may be better to add "(VV2)" after the "24.25".

Line 111: It may be better to add "(VV2)" after the "59.75".

Line 157: Please note that all Supplementary Information (Table S1, S2 and S3) are missing.

Line 257, Table 1. "Empirical 95% confidence intervals calculated from all runs are shown in brackets". All of the mean excess number are positive. It make sense. Are all of these confidence intervals values negative? If so, what does it mean?

Line 259. Thanks the authors add this Table. This table provides further information about the spatial variations of the exposure to infectious aerosol. From this table, it can be seen that Location P2 in the stalls and WM2 in the mobile grandstands are the worst locations in both ventilation versions, respectively. While, in solid grandstands the best and worst location are depend on the ventilation versions.

Line 260: It may be better to add "for Scenario 1" after the "(VV2) ".

Line 269-272: The sentence "The increase in positive cases was calculated by dividing the number of cases without MGE minus the number of cases with MGE divided by the number of cases without MGE, thus negative numbers indicate that a simulation run without MGE had higher numbers compared to the parallel run with MGE." may be need modify to "The increase in positive cases was calculated by dividing the number of cases with MGE minus the number of cases without MGE divided by the number of cases without MGE, thus negative numbers indicate that a simulation run without MGE had higher numbers compared to the parallel run with MGE."

Line 273: The same question is as above Line 257.

Line 280: Please check the value of Exit in Scenario 1, > 5 minutes, in the Figure 1b. It should be 6.4 (± 1.4).

Line 454-456: Please check the two sentences. They may be duplicated.

Line 504: Please check whether the Ref 31 should be Ref 21.

Line 537: It would be better to add little bit information about how to study scenarios including masks vs. no masks, or how to calculate the effect of wearing mask.

Line 617-619: This sentence may be not necessary.

Line: 813: Please add a full stop mark after the word "(brown)".

Line 818: The "VV1" should be "VV2".

Line 825: Thanks the authors improving this table. However, I am sorry that I am still a bit of confuse the meaning " a contact in 1st half can occur in 2nd half ".

Line 855: Please note the "with n" is still in the table.

Response to reviewers' comments to the manuscript: "The risk of indoor sports and culture events for the transmission of COVID-19"

Please note that our responses are interspaced and in bold letter

Comments to the reviewers

Reviewer #3 (Remarks to the Author):

Comments:

Thanks the authors' great effort to provide helpful responses, add some new meaningful tables and figures, as well as improve the quality of this manuscript. However, there are still some issues as following that I would like to see addressed prior to the publication of this manuscript in this high quality journal:

Line 85-84: This sentence may be not safe for the case: $>10\text{sec}$ in Scenarios 2 and 3. Because the values of "half time" (24.7, 13.3) are about 100% (83%, 118%) higher than the values of "Entry" (13.5, 6.1).

→ Thank you for pointing this out. We specified it to (line : "In Scenario 1, new contacts lasting longer than 5 minutes were created throughout the event, while in Scenarios 2 and 3 most contacts occurred during the entry phase, without further major increase (Figure 1c)."

Line 110: It may be better to add "(VV2)" after the "24.25".

Line 111: It may be better to add "(VV2)" after the "59.75".

→ In both cases (VV2) was added as suggested.

Line 157: Please note that all Supplementary Information (Table S1, S2 and S3) are missing.

→ Supplementary Information will be provided with the manuscript. It was provided upon first submission but was apparently not saved for the first revision.

Line 257, Table 1. "Empirical 95% confidence intervals calculated from all runs are shown in brackets". All of the mean excess number are positive. It make sense. Are all of these confidence intervals values negative? If so, what does it mean?

→ The values in brackets represent the range from the lower to the upper 95% confidence intervals. A hyphen is used in between both values. Only the values of the lower confidence intervals are thus really negative (represented by a minus (-) in the beginning). To make it more clear, spaces were added in front of and after the hyphen. Negative values of the lower 95% confidence interval are to be expected as the mean excess numbers are relatively close to zero and the variation is high.

Line 259. Thanks the authors add this Table. This table provides further information about the spatial variations of the exposure to infectious aerosol. From this table, it can be seen that Location P2 in the stalls and WM2 in the mobile grandstands are the worst locations in both ventilation versions, respectively. While, in solid grandstands the best and worst location are depend on the ventilation versions.

→ We are happy the table clarifies spatial variations of the exposure to infectious aerosols in the arena.

Line 260: It may be better to add "for Scenario 1" after the "(VV2)".

→ We added it. Thank you for pointing it out.

Line 269-272: The sentence “The increase in positive cases was calculated by dividing the number of cases without MGE minus the number of cases with MGE divided by the number of cases without MGE, thus negative numbers indicate that a simulation run without MGE had higher numbers compared to the parallel run with MGE.” may be need modify to “The increase in positive cases was calculated by dividing the number of cases with MGE minus the number of cases without MGE divided by the number of cases without MGE, thus negative numbers indicate that a simulation run without MGE had higher numbers compared to the parallel run with MGE.”.

→ **Thank you for finding the mistake, we have corrected accordingly.**

Line 273: The same question is as above Line 257.

→ **Please see above. Also here, spaces were added around the hyphen.**

Line 280: Please check the value of Exit in Scenario 1, > 5 minutes, in the Figure 1b. It should be 6.4 (± 1.4).

→ **Thank you very much for your thorough investigation. There was a typo that is now corrected. The correct value is 1.6 (± 1.4).**

Line 454-456: Please check the two sentences. They may be duplicated.

→ **Thank you for pointing that out. The second sentence was deleted.**

Line 504: Please check whether the Ref 31 should be Ref 21.

→ **The reference 31 is correct. However, we realized that this statement would benefit from additional references and we added the following two:**

32. Kormuth, K. A. *et al.* Environmental Persistence of Influenza Viruses Is Dependent upon Virus Type and Host Origin. *mSphere* 4, 1–14 (2019).

33. Jayaweera, M., Perera, H., Gunawardana, B. & Manatunge, J. Transmission of COVID-19 virus by droplets and aerosols: A critical review on the unresolved dichotomy. *Environmental Research* (2020) doi:10.1016/j.envres.2020.109819.

Line 537: It would be better to add little bit information about how to study scenarios including masks vs. no masks, or how to calculate the effect of wearing mask.

→ **We improved the sentence and hope it now clarifies all uncertainties. “We also studied scenarios using masks in the MGE vs. no masks. For the impact on population level, a 30-day period was studied for the outcomes. This implies that outcomes resulting from MGE were effectively counted over a shorter time period, as some late secondary and tertiary infections, as well as well as their late outcomes, might have not developed within the studied time window.”**

Line 617-619: This sentence may be not necessary.

→ **Thank you. That is true as he is now a coauthor. We deleted it.**

Line: 813: Please add a full stop mark after the word “(brown)”.

→ **This was done.**

Line 818: The “VV1” should be “VV2”.

→ **Thank you, we changed it.**

Line 825: Thanks the authors improving this table. However, I am sorry that I am still a bit of confuse the meaning “ a contact in 1st half can occur in 2nd half ”.

→ We would like to explain Extended Data Table 2 again and also provide an example: The number of cumulative contacts (which we define via time thresholds) accumulate over time, i.e. the number of contacts that were acquired in an earlier scenario section is dragged into the next section. Only new contacts of one section are added previously acquired contacts.

Using the example of the cutoff $\geq 10s$, in the scenario (see red below), the value Total = 63.9 contacts is the value of the cumulative number of contacts at the exit and corresponds to the number of contacts that a participant has accumulated on average by the end of the entire event.

The value 30.8, on the other hand, corresponds to the number of contacts that a participant only made in the entry phase, 7.0 to the contacts in 1st half, etc. Since these values correspond to all contacts that a participant has in a certain setting (independent of all other settings), and not just the newly added ones as it is the case with the cumulative contacts, it is the sum of the individual phases. I.e. a contact of a couple spending the entire event together is counted as a new contact in each setting (if a setting is considered isolated and not cumulative). In the cumulative case, this couple would only be counted once (in the entry phase) as a contact, if it already exceeds the cutoff of 10 seconds.

Time cut-off	S	Mean number of measured contacts (\pm SD)					
		Total	Entry	1 st half	half time	2 nd half	Exit
$\geq 10s$	1	63.9 (± 17.1)	30.8 (± 12.1)	7.0 (± 2.8)	24.5 (± 9.2)	6.4 (± 2.7)	15.3 (± 6.7)
	2	36.4 (± 12.0)	13.5 (± 5.3)	3.7 (± 1.5)	14.7 (± 6.7)	4.3 (± 1.8)	14.9 (± 7.6)
	3	18.0 (± 7.2)	6.1 (± 3.5)	1.3 (± 0.6)	8.3 (± 5.1)	1.8 (± 1.5)	6.1 (± 3.6)
$\geq 5min$	1	14.1 (± 5.2)	8.7 (± 4.1)	5.3 (± 2.3)	3.1 (± 2.4)	4.4 (± 2.0)	1.6 (± 1.4)
	2	6.1 (± 2.4)	4.9 (± 2.1)	2.7 (± 1.3)	2.6 (± 1.5)	3.2 (± 1.5)	1.1 (± 0.9)
	3	2.2 (± 1.5)	2.0 (± 1.3)	1.0 (± 0.3)	1.2 (± 1.9)	1.0 (± 0.7)	0.7 (± 0.7)
$\geq 15min$	1	8.9 (± 3.5)	5.1 (± 2.5)	4.5 (± 2.1)	1.8 (± 1.3)	3.9 (± 1.9)	0 (± 0)
	2	4.7 (± 1.9)	3.7 (± 1.6)	2.3 (± 1.2)	1.9 (± 1.2)	2.9 (± 1.4)	0 (± 0)
	3	1.3 (± 0.9)	1.1 (± 0.6)	1.0 (± 0.3)	0.8 (± 0.7)	0.9 (± 0.6)	0 (± 0)
$\geq 10s$ cumulative	1		30.8	32.8	52.1	54.1	63.9
	2		13.5	13.9	24.7	25.3	36.4
	3		6.1	6.3	13.3	14.1	18.0
$\geq 5min$ cumulative	1		8.7	11.0	12.3	13.7	14.1
	2		4.9	5.2	5.6	6.0	6.1
	3		2.0	2.0	2.1	2.2	2.2
$\geq 15min$ cumulative	1		5.1	7.3	7.5	8.9	8.9
	2		3.7	4.0	4.2	4.7	4.7
	3		1.1	1.1	1.3	1.3	1.3

Line 855: Please note the “with n” is still in the table.

→ Thank you. We deleted this line.